# Angiopoietin-like 2 is essential to aortic valve development in mice

Pauline Labbé [1,2✉], Victoria Munoz Goyette [1,2], Nathalie Thorin-Trescases[1], Louis Villeneuve[1], Ines Desanlis [3], Constance Delwarde [4], Yan-Fen Shi[1], Cécile Martel[1,5], Carol Yu[1,6], Azadeh Alikashani[1], Maya Mamarbachi[1], Frédéric Lesage[1,7], Samuel Mathieu[8], Jean-Claude Tardif [1,9], Patrick Mathieu [8], Marie Kmita [3] & Éric Thorin [1,2,10]

Aortic valve (AoV) abnormalities during embryogenesis are a major risk for the development of aortic valve stenosis (AVS) and cardiac events later in life. Here, we identify an unexpected role for Angiopoietin-like 2 (ANGPTL2), a pro-inflammatory protein secreted by senescent cells, in valvulogenesis. At late embryonic stage, mice knocked-down for *Angptl2* (*Angptl2*-KD) exhibit a premature thickening of AoV leaflets associated with a dysregulation of the fine balance between cell apoptosis, senescence and proliferation during AoV remodeling and a decrease in the crucial Notch signalling. These structural and molecular abnormalities lead toward spontaneous AVS with elevated trans-aortic gradient in adult mice of both sexes. Consistently, *ANGPTL2* expression is detected in human fetal semilunar valves and associated with pathways involved in cell cycle and senescence. Altogether, these findings suggest that Angptl2 is essential for valvulogenesis, and identify *Angptl2*-KD mice as an animal model to study spontaneous AVS, a disease with unmet medical need.

[1] Montreal Heart Institute, Université de Montréal, Montreal, QC, Canada. [2] Faculty of Medicine, Department of Pharmacology, Université de Montréal, Montreal, QC, Canada. [3] Faculty of Medicine, Department of Medicine, Université de Montréal, Montreal Clinical Research Institute, Montreal, QC, Canada. [4] Université de Nantes, CNRS, INSERM, l'institut du thorax, Nantes, France. [5] Mitologics, Romainville, France. [6] ThermoFisher, Boston, MA, USA. [7] École Polytechnique de Montréal, Université de Montréal, Montreal, QC, Canada. [8] Faculty of Medicine, Department of Surgery, Université Laval, Quebec, QC, Canada. [9] Faculty of Medicine, Department of Medicine, Université de Montréal, Montreal, QC, Canada. [10] Faculty of Medicine, Department of Surgery, Université de Montréal, Montreal, QC, Canada. ✉email: pauline.labbe@icm-mhi.org

Aortic valve (AoV) stenosis (AVS) is the most common valvular heart disease in developed countries[1]. Because of the lack of therapeutic options and its growing incidence, AVS is a medical condition with important societal and economic burdens[2]. Moreover, congenital malformations in valves predispose to the appearance of heart valve diseases later in life[3], and AVS represents ~5% of congenital cardiovascular defects[4]. Investigation of molecular processes involved in heart valvulogenesis could help identify key underpinnings playing a role in AVS and to develop novel therapies.

Heart valve development requires multiple molecular signals working in an integrated and time-dependent manner. In mice, it begins approximately at embryonic day 9.5 (E9.5) when a subset of endocardial cells within the atrioventricular canal (AVC) and the outflow tract (OFT)—a transient structure connecting the embryonic ventricles with the aortic sac, whose remodeling will give rise to the future AoV and pulmonary valve (PuV)—undergo epithelial-to-mesenchymal transition (EMT)[5]. Then, mesenchymal cells proliferate and invade the cardiac jelly produced by the myocardium to form the endocardial cushions[6]. After EMT, endocardial cushion cells elongate and undergo complex remodeling to give rise to the mature valve leaflets[7]. This step, from E11.5 until birth, involves valvular cell apoptosis and proliferation together with extracellular matrix (ECM) remodeling[8,9]. Recently, a study revealed the presence of intense areas of senescence, particularly in the OFT, in the developing mouse and chicken hearts[10], suggesting that embryonically programmed senescence also participates in semilunar valve remodeling.

Inflammation and tissue remodeling involving numerous growth factors contribute to AVS pathogenesis, and circulating growth factors have been recently proposed as potential biomarkers for AVS[11]. Angiopoietin like-2 (ANGPTL2) is a circulating pro-inflammatory and pro-angiogenic protein[12], a member of the large family of angiopoietin-like glycoproteins (Angptls), a family of eight (angptl1-8) members that play key roles in various biological and pathological processes. In addition to ANGPTL2, three other members of the Angptls family have been shown to be associated with cardiovascular disease (CVD): ANGPTL3, ANGPTL4 and ANGPTL8. ANGPTL3 and ANGPTL4 are main regulators of lipoprotein metabolism by inhibiting lipoprotein lipase activity, and recent reviews reported that subjects with ANGPTL3 or ANGPTL4 loss of function variants have a favorable lipid profile and a lower risk of CVD[13,14]. ANGPTL3 and ANGPTL4 may also modulate glucose metabolism and insulin sensitivity, and recent data suggest that their loss of function variants are associated with lower odds of type 2 diabetes[14]. Data concerning the role of ANGPTL8 in atherosclerosis, diabetes and obesity are more conflicting, but the clinical potential of elevated plasma ANGPTL8 to predict CVD, but also metabolic syndrome and liver disease is being explored[15]. ANGPTL2 is a member of the senescence-associated secretory phenotype[16,17] and is associated with multiple age-related chronic diseases such as cancer, diabetes, atherosclerosis and metabolic disorders[18,19]. Circulating levels of ANGPTL2 predict CVDs in the general population[20] and have been reported to be higher in patients with coronary artery disease[21–25], acute coronary syndrome[26], carotid atherosclerosis[27], heart failure[26] and dilated cardiomyopathy[28], compared to healthy subjects. ANGPTL2 expression is also upregulated in the valvular tissue of patients with calcific AVS compared to control valves[29]. However, ANGPTL2 also contributes to maintain physiological tissue homeostasis[18,30]. In rats, the injection of cardiac cells expressing Angptl2 in infarcted myocardium improved left ventricular (LV) ejection fraction (EF) and increased neovascularization[31]. ANGPTL2 promotes angiogenesis[32] and is crucial for vascular development through its antiapoptotic properties[33]. Finally, ANGPTL2 is involved in wound healing, tail and fins regeneration in amphibians and fishes[34–36],

and intestinal epithelial regeneration and bone repair in rodents[37,38]. Thus, although ANGPTL2 is secreted by senescent cells and contributes to CVDs, it is also a growth factor.

In addition to its beneficial roles in adult organisms, a few studies have revealed an unexpected role of ANGPTL2 in embryogenesis. Functional enrichment analysis[39] showed that heart development was among the top 20 functions enriched with Angptl2 co-expressed genes[40]. Niki et al. previously showed that Angptl2 was expressed in the OFT of E4 chick embryos[41]. Similarly, using single-cell RNA sequence analysis, Angptl2 was detected among other genes in mouse embryos in valve endothelial cells (VECs) during OFT remodeling at E10.5[42] and in valve interstitial cells (VICs) in mature semilunar valves at E14.5[43]. However, the role of Angptl2 in the development of the mammalian heart and valvulogenesis is largely unknown, and to the best of our knowledge, the role of the other ANGPTLs in valve biology/disease is unidentified.

In this study, we provide evidence that ANGPTL2 plays an essential role in the development of AoV. Gene and protein expression of ANGPTL2 during normal AoV development were detected in wild-type (WT) mouse embryos. We then show that knockdown of Angptl2 (Angptl2-KD) exhibited premature thickening of AoV leaflets at late embryonic stage during valve remodeling, associated with a dysregulation of cell apoptosis, senescence, proliferation, and by a decrease in Notch signalling, a key contributor of valvulogenesis. Adult Angptl2-KD mice, from the age of 2 to 10 months, develop AVS characterized by collagen disorganization and elevated trans-aortic gradient, without developing severe LV dysfunction or heart failure. In agreement with these data in mice, the expression of ANGPTL2 and NOTCH1 was elevated in human fetal semilunar valves and associated with pathways related to cell cycle and senescence. Taken together, these findings identify a crucial role for ANGPTL2 in AoV remodeling during heart valvulogenesis.

## Results

**ANGPTL2 is expressed at early stages of AoV development in WT mice.** In situ hybridization with Angptl2 RNA probes on whole WT embryos revealed the expression of Angptl2 at early stages of heart valve development. At E9.5 (Fig. 1a) and E10 (Fig. 1b), low expression of Angptl2 was found in the developing brain, as previously observed in chick embryos[41]; In addition, Angptl2 expression was found in the developing forelimbs and hindlimbs and the branchial arches. Using our previously described Angptl2-KD mice[44,45], we validated the specificity of the Angptl2 RNA probe: control in situ hybridizations on Angptl2-KD embryos at E9.5 (Fig. 1e) and E10.5 (Fig. 1f) provided no signal. At E11 (Fig. 1c) and E11.5 (Fig. 1d), Angptl2 expression increased in the previously mentioned areas, and a specific signal was observed in the OFT (Fig. 1g, h), as reported in the chick embryo[41]. In accordance with our results, in situ hybridization of WT C57Bl/6 mouse embryos at E14.5 from the public database Eurexpress (www.eurexpress.org)[46] shows expression of Angptl2 RNA in all four cardiac valves (Fig. S1).

In addition, the spatial expression of ANGPTL2 in AoV leaflets during its development was assessed by immunofluorescence in cardiac histological sections from E14.5 and E18 WT mouse embryos. At E14.5, ANGPTL2 was expressed at the junction of the three leaflets of the PuV and the AoV (Fig. 1i). Higher magnifications show that ANGPTL2 and CD31 are co-expressed in the cytoplasm of VECs in both the PuV (Fig. 1i; 1) and the AoV (Fig. 1i; 2). At E18, ANGPTL2 expression was also restricted to the border of the leaflet in the AoV (Fig. 1j).

Similarly, in 2-month old WT mice, ANGPTL2 was strongly and specifically expressed in the AoV leaflets (Fig. 1k; panels 1, 2)

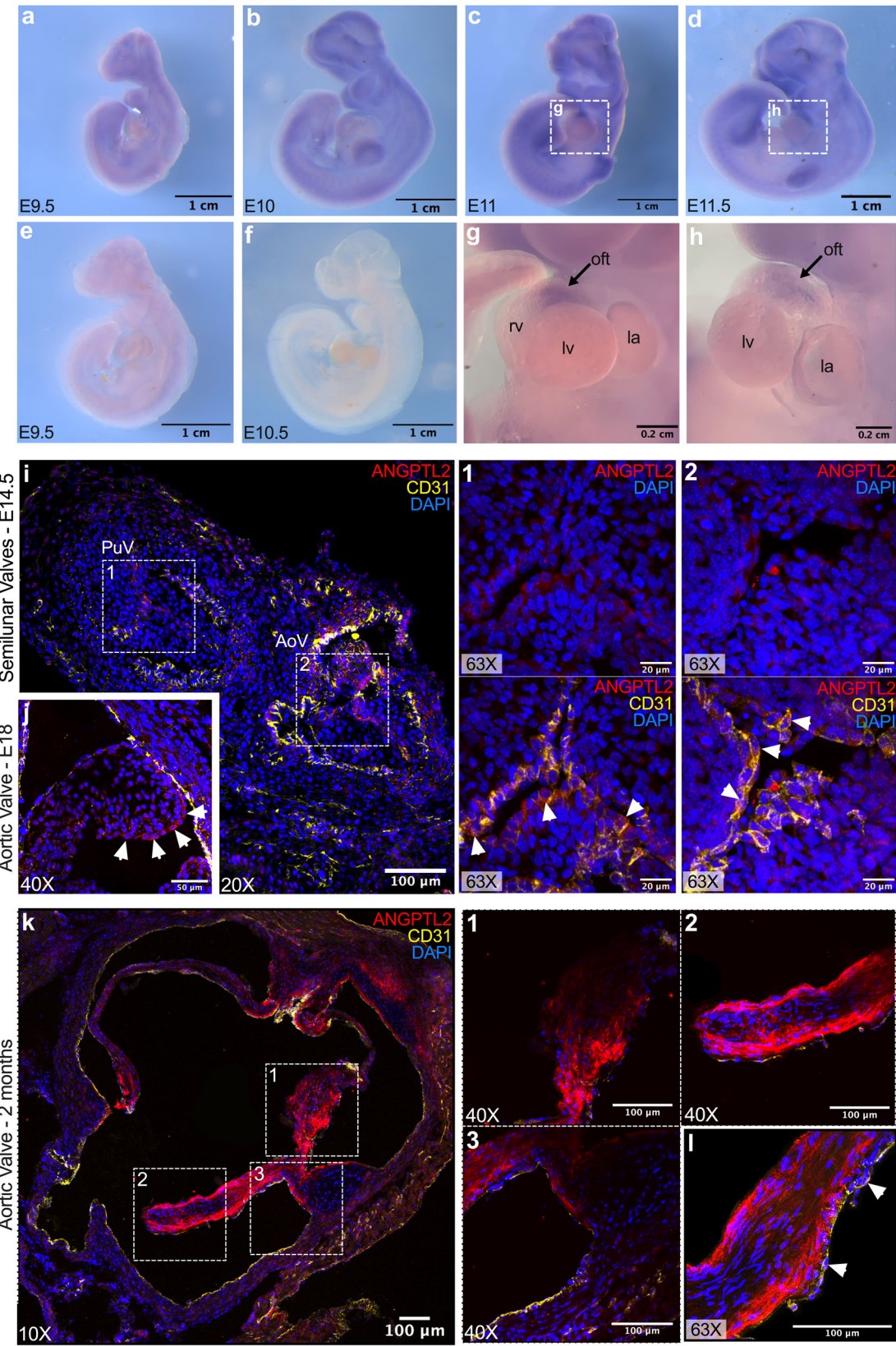

rather than in the aortic root and the leaflet attachments (Fig. 1k; panel 3). Higher magnification shows that ANGPTL2 was mostly expressed at the edge of the leaflet, in accordance with what was observed at E14.5 (Fig. 1i) and E18 (Fig. 1j) in WT mice, and also in VICs rather than in the CD31-positive VECs (Fig. 1l). Western Blot experiments confirmed that ANGPTL2 is expressed in VICs

sub-cultured from AoV leaflets of 2-month old WT mice (Fig. S2a; lines 1–2; Fig. S2b; line 1) compared to a positive control, i.e., HEK293 cells transfected with a plasmid encoding the murine form of ANGPTL2 (Fig. S2a; line 3; Fig. S2b; lines 2–3). We verified that our isolated valve cell population was not contaminated by endothelial cells. Indeed, isolated valve cells did

**Fig. 1 ANGPTL2 is expressed at early stages of AoV development in WT mice. a–d** Representative views of in situ hybridizations with *Angptl2* RNA probes on WT mice embryos at E9.5, E10, E11 and E11.5. **e, f** Representative views of in situ hybridizations with *Angptl2* RNA probes on *Angptl2*-KD mice embryos at E9.5 and E10.5. **g, h** Higher magnifications of the heart region at E11 and E11.5; The outflow tract (oft) is indicated by black arrows. **i, j** Representative ANGPTL2 and CD31 protein expression by immunofluorescence in cardiac sections from (**i**) E14.5 and from (**j**) E18 embryo from WT mice. At a higher magnification (×63), white arrows show the expression of ANGPTL2 on the edge of the leaflet and indicate cells co-expressing ANGPTL2 and CD31 in E14.5 embryo. **k** Representative ANGPTL2 and CD31 protein expression by immunofluorescence in cardiac sections from 2-month old WT mice. **l** Higher magnification (×63) of ANGPTL2 and CD31 expression in AoV leaflets of WT mice: ANGPTL2 is not co-expressed with CD31-positive cells (indicated by white arrows). All images are representative of n = 3 independent experiments. DAPI staining was used to visualize cell nuclei.

not express the endothelial marker VE-cadherin (Fig. S2b, line 1), unlike the positive control HUVECs (Fig. S2b, lines 4–5), but rather expressed markers of (myo)fibroblasts characteristic of VICs such as vimentin and α-SMA (Fig. S2b, line 1). The expression of vimentin and α-SMA in our population of VICs was also confirmed by immunofluorescence (Fig. S2c). The cytoplasmic expression of ANGPTL2 was verified in cultured VICs from WT mice by immunofluorescence (Fig. S2d). Of note, the faint nuclear staining observed in cultured VICs from adult *Angptl2*-KD mice, but also in cardiac histological sections from E14.5 WT and *Angptl2*-KD mice is a non-specific signal (Fig. S3).

Altogether, these results suggest that during AoV formation in WT mice, ANGPTL2 is expressed from E11 to young adult stage, with its expression restricted to the VECs at the edge of the leaflet at embryonic stage, and then extended to the VICs at the border of the leaflet at adult stage.

**Angptl2-KD mice exhibit AoV defects at embryonic stage.** We then examined the impact of knocking down *Angptl2* on AoV development, using *Angptl2*-KD mice[44,45]. *Angptl2*-KD mouse embryos at E9.5 and E10.5 did not express *Angptl2* (Fig. 1e, f), and *Angptl2* mRNA expression in AoV leaflets was also undetectable in adult *Angptl2*-KD mice (Fig. 2a). Examination of dissected *Angptl2*-KD hearts at E10.5 (Fig. 2b) and E11.5 (Fig. 2c) did not reveal notable difference in the size of the OFT compared to age-matched embryos from WT mice. Likewise, heart sections from *Angptl2*-KD mice embryos at E14.5 stained with hematoxylin and eosin did not show any difference in AoV leaflets thickness compared to WT (Fig. 2d).

In contrast, heart sections from *Angptl2*-KD mice embryos at E18 showed an enlargement of both the leaflets (Fig. 2e) and hinges (Fig. 2f) of the AoV, compared to age-matched WT embryos. Measurements of the AoV thickness demonstrated early thickening of both the leaflets (from $73.5 \pm 3.9$ µm in WT to $90.9 \pm 7.0$ µm in *Angptl2*-KD, $p < 0.05$; Fig. 2g) and hinges (from $37.9 \pm 1.6$ µm in WT to $49.2 \pm 3.1$ µm in *Angptl2*-KD, $p < 0.01$; Fig. 2h). Hence, these data strongly suggest that ANGPTL2 is involved in the remodeling of fetal AoV.

**AoV defects in Angptl2-KD mice are associated with an impairment in the balance between embryonic apoptosis, senescence and proliferation.** We then explored the cellular mechanisms responsible for the development of premature AoV leaflet thickening in *Angptl2*-KD embryos. We hypothesized that ANGPTL2 could regulate programmed embryonic apoptosis, senescence and/or proliferation during heart valve remodeling. TUNEL assay for detection of apoptosis and immunofluorescence for the senescence marker p21 and the proliferation marker Ki67 were performed on heart sections from WT (Fig. 3a) and *Angptl2*-KD (Fig. 3b, c) mouse embryos at E14.5. The analysis of TUNEL, p21 and Ki67-positive cells specifically in the AoV leaflets (Fig. 3d–f) showed an imbalance between senescence/apoptosis and proliferation in *Angptl2*-KD mice compared to WT mice: pie charts illustrating the percentage of marked cells of the total nuclei, and the corresponding proportion (%) of TUNEL,

p21 and Ki67-positive cells within the marked cells (Fig. 3g, h and i) show (i) high proliferation in *Angptl2*-KD, with heterogeneous increased number of Ki67-positive cells between AoV of *Angptl2*-KD embryos and (ii) low senescence and apoptosis in AoV leaflets from *Angptl2*-KD embryos. Indeed, in AoV from *Angptl2*-KD embryo with severe valve thickening (Fig. 3b'), the level of proliferation is high with 97.6% Ki67-positive cells (Fig. 3e–h) and is associated with low cell death/senescence with 2.4% TUNEL + p21-positive cells of the total marked cells (Fig. 3h). On the other hand, in AoV from *Angptl2*-KD embryo with thinner leaflets (Fig. 3c'), the level of proliferation is moderate with 37% Ki67-positive cells (Fig. 3f–i) and is associated with higher cell death/senescence with 63% TUNEL + p21-positive cells of the total marked cells (Fig. 3i). In AoV from WT mice, Ki67-positive cells represent only 4% (Fig. 3d–g) while TUNEL + p21-positive cells represent 96% of the total marked cells (Fig. 3g). Overall, in AoV from WT embryos, the proportion of TUNEL + p21-positive cells is the highest, representing an average of 6.1% (n = 4 embryos) when expressed as a % of the total nuclei number. In contrast, in AoV from *Angptl2*-KD embryos, the proportion of TUNEL + p21-positive cells is lower, representing an average of 2.8% (n = 4 embryos) of the total nuclei number.

In average, we observed both a significant decrease in apoptotic cells (−47%, $p < 0.05$; Fig. 3j) and in the number of senescent cells expressing p21 (−48%, $p < 0.05$; Fig. 3j), and a tendency toward an increase in the percentage of cells expressing the proliferation marker Ki67 (Fig. 3j, +396%, $p = 0.13$) in AoV from *Angptl2*-KD embryos compared to WT embryos.

Interestingly, we observed that p21 was mostly expressed in VECs (as p21 was co-expressed with CD31); in addition, co-staining of ANGPTL2 and p21 in WT embryos showed that they are co-expressed in AoV (Fig. 3k) sections. Higher magnification confirmed the presence of ANGPTL2 and p21 in endothelial cells (Fig. 3k), suggesting a role for ANGPTL2 in embryonic senescence during OFT remodeling and valve development.

These results strongly suggest that AoV leaflets from *Angptl2*-KD mice are prone to reduced apoptosis and senescence, and increased proliferation at E14.5—the lower the senescence and apoptosis, the higher the proliferation—leading to thickened AoV leaflets at E18.

**Adult Angptl2-KD mice develop spontaneous AVS.** We then examined the impact of the knockdown of *Angptl2* in 2-month old mice, with a further follow-up at 7–10 months. In accordance with the premature structural abnormalities observed at E18 during AoV development in *Angptl2*-KD embryos (Fig. 2e–h), Masson's trichrome staining of heart sections from 2-month old male and female *Angptl2*-KD mice showed the enlargement of both the leaflets (Fig. 4a) and hinges (Fig. 4b) of the AoV, when compared to age- and sex-matched WT mice. Quantification of valve thickness shows the enlargement of both the leaflets (from $90.9 \pm 7.6$ µm in WT to $113.0 \pm 7.7$ µm in *Angptl2*-KD, $p < 0.05$; Fig. 4c) and hinges (from $20.4 \pm 1.1$ µm in WT to $39.0 \pm 3.6$ µm in *Angptl2*-KD, $p < 0.0001$; Fig. 4d) of the AoV of male and female *Angptl2*-KD mice compared to WT mice.

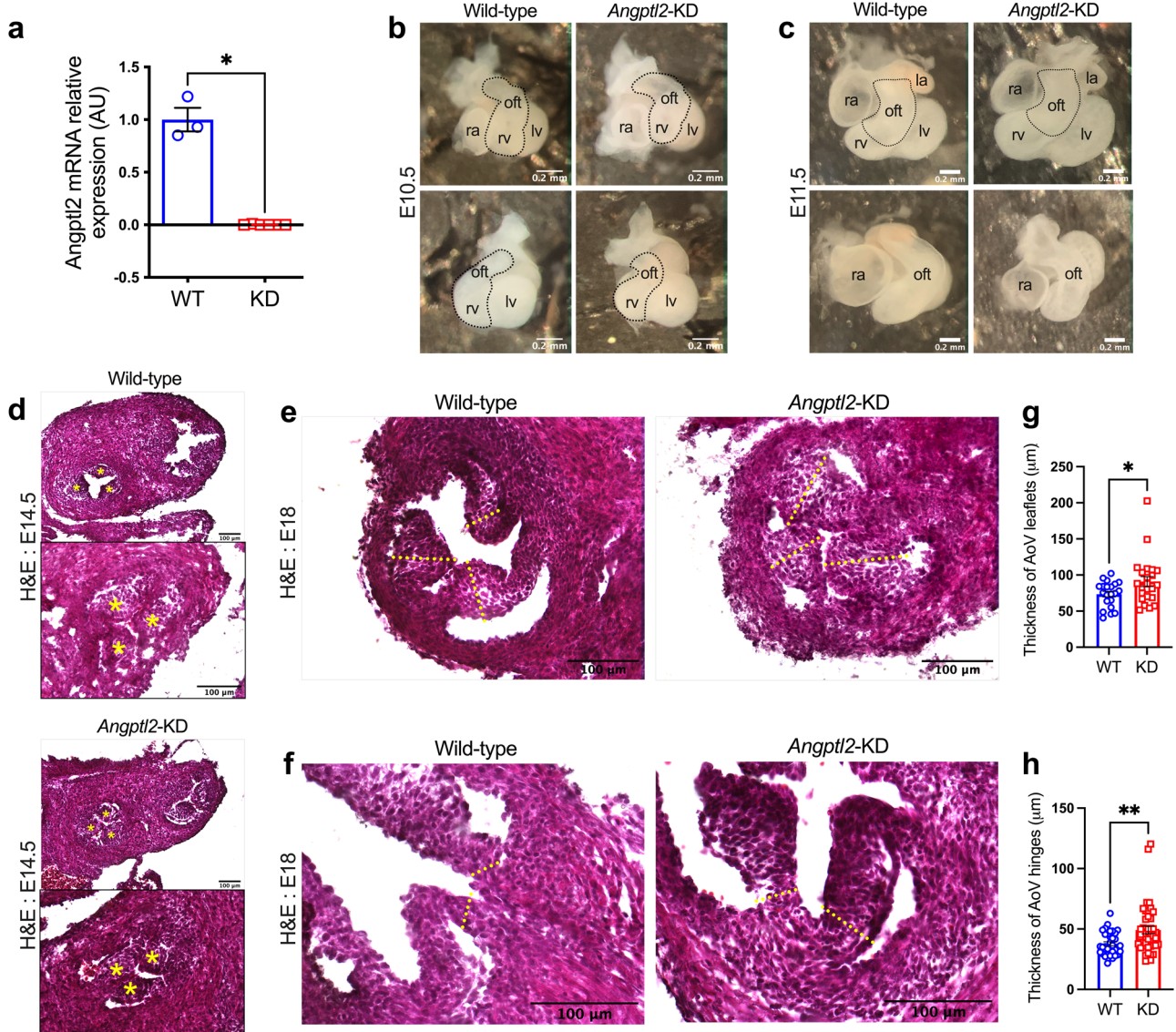

**Fig. 2 _Angptl2_-KD mice exhibit AoV defects at embryonic stage. a** _Angptl2_ gene expression measured by quantitative RT-qPCR in AoV leaflets from adult male WT and _Angptl2_-KD mice. Data are mean ± SEM of $n = 3$ for WT and $n = 6$ for _Angptl2_-KD mice; *: $p < 0.05$ determined with Mann–Whitney U test. **b**, **c** Front and side views of representative hearts from E10.5 and E11.5 embryos from WT and _Angptl2_-KD mice. Images are representative of $n = 3$ independent experiments. **d** Representative heart sections from E14.5 embryos from WT and _Angptl2_-KD mice stained with Hematoxylin and Eosin. Higher magnification of AoV at E14.5 is shown in the bottom panel; the 3 leaflets of AoV are indicated by yellow stars. Images are representative of $n = 3$ independent experiments. **e**, **f** Representative AoV sections from E18 embryos from WT and _Angptl2_-KD mice stained with Hematoxylin and Eosin. The yellow dotted lines represent the regions used for quantification. Quantification of valve thickness shows the enlargement of both (**g**) the leaflets and (**h**) the hinges of AoV in E18 embryos from _Angptl2_-KD mice compared to WT. Data are means ± SEM of $n = 8$ for WT and $n = 9$ for _Angptl2_-KD mice; up to 3 leaflets ($n = 21$ for WT and $n = 23$ for _Angptl2_-KD mice) and 6 hinges ($n = 34$ for WT and $n = 42$ for _Angptl2_-KD mice) were quantified _per_ section; *: $p < 0.05$ and **: $p < 0.01$ determined with unpaired $t$-test.

Picrosirius red staining of the AoV of male and female _Angptl2_-KD mice revealed disorganized and less tightly packed collagen fibers, when compared to WT mice (Fig. 4e). Moreover, the analysis of collagen maturity under polarized light showed the presence of more yellow-green fibers characteristic of loosely packed and less cross-linked collagen fibers in _Angptl2_-KD AoV, rather than orange-red fibers characteristic of tightly packed and organized collagen fibers in WT AoV (Fig. 4f). Histological alcian blue, alizarin red and oil red staining showed no proteoglycan remodeling, no calcification and no lipid deposition in AoV, respectively, in 2-month old male _Angptl2_-KD mice (Fig. 4g).

Finally, we performed an echocardiography analysis of 2-month old _Angptl2_-KD mice to further evaluate AVS. Echocardiography showed that 80% of the _Angptl2_-KD mice, of both sexes, exhibited spontaneous AVS with significantly increased trans-aortic velocity compared to the normal value of 90-150 cm/sec in mice[47] and thickened AoV leaflets, when compared to age- and sex-matched WT littermates (Fig. 4h). The mean pressure gradient, peak aortic jet velocity and leaflet thickness of _Angptl2_-KD mice were significantly increased, while the valve area was significantly reduced when compared to WT mice (Fig. 4h). At 2 months, echocardiographic data showed no notable LV systolic and diastolic dysfunction (Table S1). Pulsed-wave Doppler velocity tracings and 2D imaging of AoV (Fig. 4i) also illustrate the increased trans-valvular AoV velocity and leaflet thickness in 2-month old _Angptl2_-KD male mice compared to WT littermates.

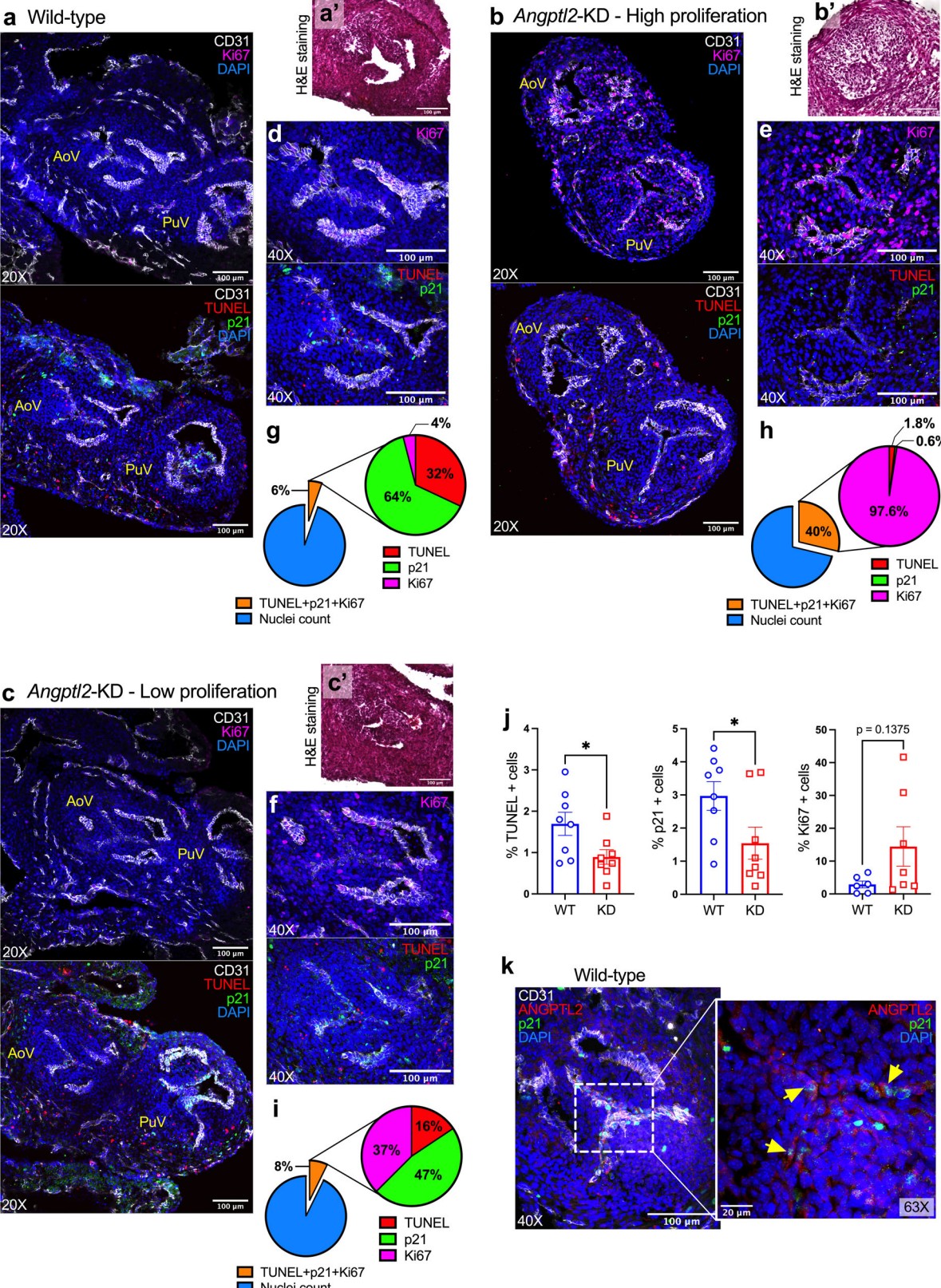

Altogether, these data show that *Angptl2*-KD mice of both sexes exhibit AVS. Of note, at the age of 2 months, although trans-mitral valve (MV) peak velocity and mean gradient were significantly higher in *Angptl2*-KD mice than in WT mice, values were all clinically normal in the 20 mice scanned (Fig. S4), indicating that there was no substantial remodeling.

AVS was still present in 3.5–7-month old male *Angptl2*-KD mice, as confirmed by echocardiography (Fig. 4j) and optical coherence tomography (Fig. S5a). The severity of AVS was similar in 2- and 7-month old *Angptl2*-KD mice, suggesting that the stenosis is established early and is not transient (Fig. 4j). Moreover, the distribution of AVS phenotype severity (evaluated

**Fig. 3 AoV defects in *Angptl2*-KD mouse embryos are associated with an impairment in the balance between embryonic apoptosis, senescence and proliferation.** Representative images (of $n = 3$ independent experiments) of TUNEL assay and Ki67, p21 and CD31 protein expression by immunofluorescence in cardiac sections from E14.5 embryos from WT (**a**) and *Angptl2*-KD mice (**b**, example of High proliferation; **c** example of Low proliferation), showing both aortic valve (AoV) and pulmonary valve (PuV). In the same WT or *Angptl2*-KD embryo AoV section, H&E staining (**a′-b′-c′**) and its corresponding TUNEL assay, Ki67 and p21 protein expression (**d-f**) are shown. Pie charts illustrate the percentage of marked cells of the total nuclei, and the corresponding proportion (%) of TUNEL, p21 and Ki67-positive cells within the marked cells (**g-i**). **j** For TUNEL/p21: The graphs represent means ± SEM of $n = 4$ embryos *per* genotype (2 sections analyzed/embryo; *: $p < 0.05$ determined with unpaired *t*-test); For Ki67: The graph represents means ± SEM of $n = 6$ embryos for WT and $n = 7$ embryos for *Angptl2*-KD (1 section analyzed/embryo; $p = 0.1375$ determined with Mann–Whitney U test). **k** Representative images (of $n = 3$ independent experiments) of ANGPTL2 and p21 protein expression by immunofluorescence in AoV from E14.5 embryos from WT mice. Higher magnification (×63) shows cells co-expressing ANGPTL2 and p21 (indicated by yellow arrows). DAPI staining was used to visualize cell nuclei.

according to trans-valvular aortic velocity values) from 2 to 7 months of age (Fig. 4j–k) shows that the stenosis observed in *Angptl2*-KD mice is mild to moderate, and did not worsen with age: before 6 months, 60% of the *Angptl2*-KD mice developed mild AVS (Vmax ≥ 150 cm/s) and 13% developed moderate AVS (Vmax ≥ 300 cm/s); after 6 months, 74% of the *Angptl2*-KD mice developed mild AVS, and 4% developed moderate AVS. Male and female 7-month old *Angptl2*-KD mice showed only minimal calcification in the aortic root and the leaflet attachment where calcification often begins (Fig. S5b). Accordingly, the expression of genes involved in calcification (*Spp1*, *Alp1*) and those encoding the components of valve layers spongiosa (*Dcn* and *Vcan*) and elastis (*Eln*) were not affected in freshly isolated AoV leaflets from 10-month old male *Angptl2*-KD mice when compared to age- and sex-matched WT mice (Fig. S5c). In contrast, *Col1a1*, *Col3a1* and *Mmp2* expression decreased (Fig. S5c), supporting ECM disorganization and remodeling in adult *Angptl2*-KD AoV leaflets. Altogether, our data suggest that adult (from 2 to 10 months of age) *Angptl2*-KD mice of both sexes develop precocious spontaneous AVS with collagen disorganization and remodeling mostly at the base of the AoV leaflets, with no or minimal calcification.

At 7 months, compared to male WT littermates, *Angptl2*-KD mice showed the onset of mild LV dysfunction, with the EF decreasing by 11% but remaining within the normal range (74.6 ± 1.7 vs. 66.4 ± 3.1%; $p < 0.05$); there were also reductions in fractional shortening (FS; 38.2 ± 1.4 vs. 32.7 ± 2.4%; $p = 0.0583$), lateral contractility (2.40 ± 0.09 vs. 1.87 ± 0.08 cm/s; $p < 0.001$) and septal contractility (2.55 ± 0.09 vs. 2.18 ± 0.08 cm/s; $p < 0.01$; Table S2). Diastolic function was impaired in male *Angptl2*-KD mice compared to WT littermates, with the E/E' ratio increasing by 70% (26.2 ± 2.2 vs. 43.4 ± 4.1; $p < 0.01$; Table S2).

We did not find any difference in the levels of *Nppa* ($p = 0.5358$; $n = 7–8$ mice *per* genotype) and *Nppb* ($p = 0.7789$; $n = 7–8$ mice per genotype) in hearts of *Angptl2*-KD mice compared to WT littermates, suggesting that severe LV remodelling was not occurring at 7 months, in accordance with the mild LV dysfunction observed at that age (Table S2). Of note, in *Angptl2*-KD mice, the trans-valvular aortic Vmax (Fig. S6a) and mean pressure gradient (Fig. S6b) were positively correlated to LV mass ($r = 0.8440$, $r = 0.8938$, respectively; $p < 0.01$), LV mass/ LV dimension at end-cardiac diastole (LVDd; $r = 0.8477$, $r = 0.8349$, respectively; $p < 0.01$), and tended to be associated to LV mass/body weight (BW; $r = 0.5984$, $r = 0.6802$, respectively; $p ≤ 0.1171$). In other terms, in *Angptl2*-KD mice, the severity of AVS was correlated to the magnitude of LV hypertrophy. Similarly, heart expression of the cardiac hypertrophy marker *Myh7* was negatively correlated with aortic valve area, and positively correlated with Vmax and mean pressure gradient ($r = -0.7106$, $r = 0.7610$ and $r = 0.8201$, respectively; $p < 0.05$; Fig. S6c). Moreover, *Myh7* expression tended to be positively correlated to LV mass ($r = 0.7039$, $p = 0.0513$) and to LV mass/LVDd ($r = 0.6984$, $p = 0.054$; Fig. S6d), and was

positively associated with LV mass/BW ($r = 0.7816$; $p < 0.05$; Fig. S6d).

Finally, we evaluated the lifespan of *Angptl2*-KD mice compared to WT littermates: at 1- and 1.5-year-old, *Angptl2*-KD and WT littermates had the same survival rate: *Angptl2*-KD mice did not show a decrease of lifespan when compared to WT mice (Fig. S7).

Taken together, these results indicate that LV function is only slightly impaired in 7-month old *Angptl2*-KD mice compared to WT littermates. Although most *Angptl2*-KD mice present mild-to-moderate AVS, ~5% of animals develop a more severe AoV phenotype and are more susceptible to developing cardiac hypertrophy and heart failure. Overall, the longevity of *Angptl2*-KD mice was unaltered compared to WT littermates.

**AoV defects in *Angptl2*-KD mice are associated with reduced Notch1 signalling.** We subsequently investigated which signalling molecular pathways were responsible for the AVS observed in *Angptl2*-KD mice, at both embryonic and adult stages. *NOTCH1* has been repeatedly identified as a crucial contributor of heart valve remodeling by regulating proliferation and apoptosis of valvular cells during embryogenesis[48], and has been found to be involved in multiple congenital heart valve diseases such as bicuspid AoV (BAV) or calcific AoV disease in both human and animal models[49–51]. We, therefore, performed Notch1 immunofluorescence staining on heart sections from WT and *Angptl2*-KD mice embryos at E14.5 (Fig. 5a, b): the analysis, focused on the AoV leaflets, showed a significant decrease in Notch1 signal (Fig. 5c, −47%, $p < 0.01$) in *Angptl2*-KD mice compared to WT mice. In 2-month old mice, the reduction of activated Notch1 (Act-Notch1) expression was also observed at the protein level by immunofluorescence on cardiac histological sections (Fig. 5d; −28%; $p < 0.01$). Immunofluorescence for ANGPTL2 and Notch1 in AoV sections from WT mice embryo at 14.5 and from adult 2-month old WT mice (Fig. 5e) revealed the presence of cells co-expressing ANGPTL2 and Notch1, suggesting that both proteins could be involved in the regulation of crucial cell processes for heart valve development and homeostasis.

Analysis of gene expression in freshly isolated AoV leaflets from 4-month old male and female *Angptl2*-KD mice also showed a decrease (−39%, $p < 0.05$) in *Notch1* signalling, including its downstream targets *Hey1* and *Hey2* (−42%, $p < 0.05$ and −43%, $p < 0.01$, respectively; Fig. 5f). Interestingly, the decrease in *Notch1*, *Hey1* and *Hey2* mRNA expressions were strongly positively correlated to the decrease of AoV area ($r = 0.6886$, $r = 0.7046$, $r = 0.7724$, respectively, $p < 0.01$; Fig. 5g) and negatively correlated to the increase of AoV leaflet thickness ($r = -0.5775$, $r = -0.5626$, $r = -0.5682$, respectively, $p < 0.05$; Fig. 5h), i.e., the reduction in Notch1 observed in *Angptl2*-KD mice was proportional to the severity of AVS. Expression of *Axin2*, a downstream target and an inhibitor of Wnt/β-catenin pathway, was also significantly reduced (−30%, $p < 0.05$; Fig. S8) in *Angptl2*-KD mice. In contrast, the expression of the gene

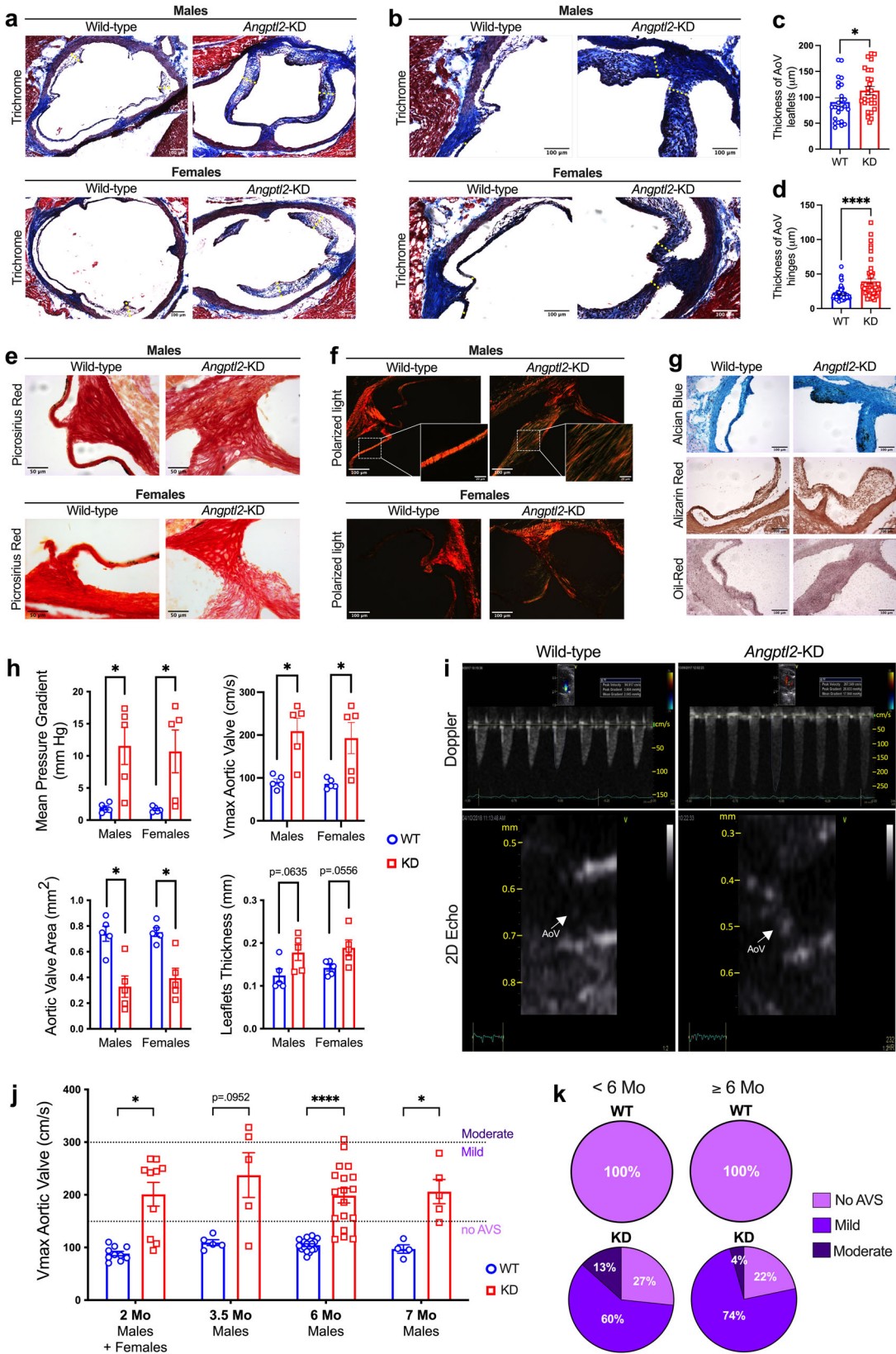

encoding β-catenin (*Ctnnb1*) was not changed. Of note, a previous study showed that *Axin2*-KO mice develop AVS with thickened leaflets characterized by a disorganization of collagen fibers, absence of calcification in older mice[52]. Valve leaflets from *Axin2*-KO mice exhibit increased black deposits, identified as melanin secreted by melanocytes[52], a feature also observed in

2-month old *Angptl2*-KD mice (Fig. S9). Interestingly, the presence of pigmented cells has been previously reported in heart valve remodelling, but their functions are not known[53]. The expression of *NF-κB*, *TGF-β1* and *Sox9*, which are modulated in calcific AVS[54–56], were not altered in *Angptl2*-KD mice compared to WT littermates (Fig. S8). However, the expression of *Bmp2* was

**Fig. 4 Adult *Angptl2*-KD mice develop spontaneous AVS. a, b** Representative AoV sections (of $n = 11$–13 independent experiments *per* genotype) from 2-month old male and female WT and *Angptl2*-KD mice stained using Masson's Trichrome. The yellow dotted lines show the regions used for the quantification. Quantification of valve thickness shows the enlargement of both (**c**) the leaflets and (**d**) the hinges of AoV in male and female *Angptl2*-KD mice compared to WT. Data are means ± SEM of $n = 13$ for WT mice, 7 males and 6 females; $n = 11$ for Angptl2-KD mice, 5 males and 6 females; up to 3 leaflets ($n = 25$ for WT and $n = 27$ for *Angptl2*-KD mice) and 6 hinges ($n = 64$ for WT and $n = 51$ for *Angptl2*-KD mice) were quantified *per* section. *: $p < 0.05$ determined with unpaired *t*-test. **e** Representative AoV sections (of $n = 3$ independent experiments) from 2-month old male and female WT and *Angptl2*-KD mice stained using Picrosirius red under white light and (**f**) Picrosirius red under polarized light. **g** Representative AoV sections (of $n = 3$ independent experiments) from 2-month old male WT and *Angptl2*-KD mice stained using alcian blue, alizarin red and oil-red. **h** Echocardiographic measurements of the mean pressure gradient, the peak aortic jet velocity (Vmax), the leaflets thickness and the AoV area in male ($n = 5$) and female ($n = 5$) WT and *Angptl2*-KD mice at 2 month-old ($n = 10$ mice *per* genotype). Data are means ± SEM of n mice. *: $p < 0.05$ determined with Mann–Whitney U test. **i** Pulsed-wave Doppler velocity tracings illustrating the increased cross AoV velocity in an *Angptl2*-KD male mouse compared to a WT littermate and 2D images of AoV (indicated by white arrows) in the long-axis view from WT and *Angptl2*-KD mice. **j** Echocardiographic measurements of the maximal trans-aortic velocity (Vmax) in *Angptl2*-KD mice compared to WT littermates from 2 months to 7 months. Data are means ± SEM of: $n = 10$ mice *per* genotype at 2 months; $n = 5$ mice *per* genotype at 3.5 months; $n = 18$ mice *per* genotype at 6 months; $n = 4$ WT mice and $n = 5$ *Angptl2*-KD mice at 7 months. *: $p < 0.05$ determined with unpaired *t*-test or Mann–Whitney U test according to the normality of distribution. **k** Pie charts illustrating the repartition of the phenotype severity for the AVS (defined according to Vmax: a Vmax ≥150 cm/s corresponds to a mild AVS, while a Vmax ≥300 cm/s corresponds to a moderate AVS) in young (<6-month-old) and older (≥6-month-old) WT and *Angptl2*-KD mice.

strongly increased ($+131\%$, $p < 0.01$) and the expression of *Bmp4* was decreased ($-31\%$, $p < 0.05$) in *Angptl2*-KD mice (Fig. S8).

To take a step further in the elucidation for the potential mechanism of ANGPTL2-dependent Notch activation in valve cells, we performed immunofluorescence in AoV sections from embryonic (E14.5) and adult (2-month old) WT mice, for the two most well-known potential receptors for ANGPTL2: (i) PirB, the mouse homolog for LILRB2, and (ii) Integrin α5β1. At E14.5, Integrin α5β1 is expressed in the membrane of cells which are localized preferably in the border of AoV leaflet, where it is found co-expressed with ANGPTL2 and Notch1 (Fig. 5i). In contrast, PirB is not expressed in embryonic AoV or in PuV (Fig. S10a); PirB is expressed only in few cells with polylobed nucleus, characteristic of neutrophils, scattered across the valve leaflets (Fig. S10b). In 2-month-old adult mice, however, PirB is sparsely expressed on the border of the leaflet, and is co-expressed in some cells expressing ANGPTL2 (Fig. 5j). At 2 months, Integrin α5β1 is not expressed in the leaflets (Fig. S10c), but it is rather expressed in the pillar of the valve, and thus no longer co-expressed with ANGPTL2 (Fig. S10d). These results suggest that embryonic ANGPTL2 signalling at E14.5 acts *via* the Integrin α5β1 rather than *via* PirB receptors. ANGPTL2/Integrin α5β1 are expressed together with Notch1 in VECs, at the border of the developing leaflet.

In summary, these results reveal that AoV leaflets from *Angptl2*-KD mice are characterized by a decrease in Notch signalling, a key regulator of heart valve development and maintenance, and a deregulation of the fine balance between apoptosis, senescence and proliferation of valve cells at E14.5, leading to thickened AoV leaflets at E18 and further AVS in adult male and female mice (Fig. 5k).

**ANGPTL2 is enriched in human fetal semilunar valves and associated with cell cycle and senescence pathways.** Finally, we evaluated the expression of *ANGPTL2* in human fetal and adult AoV by leveraging array data obtained in 16 valves (GSE45821)[57]. Expression in 8 adult AoV (aged 19–55 years/old) was compared to fetal semilunar valves (gestational age 9–17 weeks). Genes with a fold-change >1.5 at a false discovery rate >5% (FDR < 0.05) were considered differentially expressed. In total, 2918 genes were upregulated in fetal semilunar valves, whereas 3028 genes were upregulated in adult AoV (Supplementary Data 1; Fig. 6a). The expression of *ANGPTL2* (log2 fold-change 1.9, FDR = 1.40E−07) was significantly upregulated in fetal semilunar valves. Also, the expression of *HEY1* (log2 fold-change 1.8, FDR = 1.55E−06) and *NOTCH1* (log2 fold-change 1.7, FDR = 3.52E−06), two genes involved in cell fate and developmental pathways relevant to heart

valve in human, were enriched in fetal semilunar valves (Fig. 6b). In order to probe gene module expression and infer function, we carried out a weighted gene co-expression network analysis (WGCNA) by using gene expression retrieved from the human fetal and adult valves. WGCNA analysis identified 17 gene modules (Fig. S11). *ANGPTL2* was a component of the turquoise gene module, which also included *CDKN1A* (p21), *TP53*, *HEY1* and *WNT3* (Supplementary Data 2). A pathway analysis of the turquoise gene module using the REACTOME dataset showed enrichment in cell cycle, signalling by Rho GTPase and cellular senescence (Supplementary Data 3; Fig. 6c). Taken together, these data suggest that *ANGPTL2* is enriched in human fetal semilunar valves compared to valves from young and middle-aged adults, and is associated with a heart valve gene co-expression module relevant to cell cycle and senescence.

## Discussion

Aortic valve (AoV) abnormalities during embryogenesis are a major risk of AVS and cardiac pathological outcomes later in life. The mechanisms involved in the development of AVS remain, however, poorly understood as few animal models of spontaneous heart valve diseases are available. In the present study, we identify *ANGPTL2* as a potentially essential player in valvulogenesis, and describe another animal model of spontaneous valvulopathy. We found that the knockdown of *Angptl2* altered Notch signalling and disturbed the embryonic balance between cell apoptosis, senescence and proliferation in AoV leaflets during valve remodelling. The latter events were associated with the development of thickened AoV leaflets during embryogenesis and mild-to-moderate AVS in adult mice, stenosis that does not translate into severe cardiac dysfunction and heart failure. Moreover, we observed that *ANGPTL2* is enriched in human fetal semilunar valves compared to adult AoV and is associated with pathways related to cell cycle and senescence. Altogether, this study militates for a key role of *ANGPTL2* in heart valvulogenesis, and embryonic disruption of its activity may contribute to AVS in adults (Fig. 7).

During heart development in mice, valve formation begins with endocardial cells within the AVC and the OFT regions undergoing EMT and invading the cardiac jelly secreted by the underlying myocardium to form the endocardial cushions[5,6]. After EMT and from E11.5 to birth, endocardial cushion cells elongate and undergo complex remodelling to give rise to the mature valve leaflets[7]. Considering the very low expression of *Angptl2* before E11 in mouse embryos, we propose that ANGPTL2 might be involved in heart valve remodelling, rather

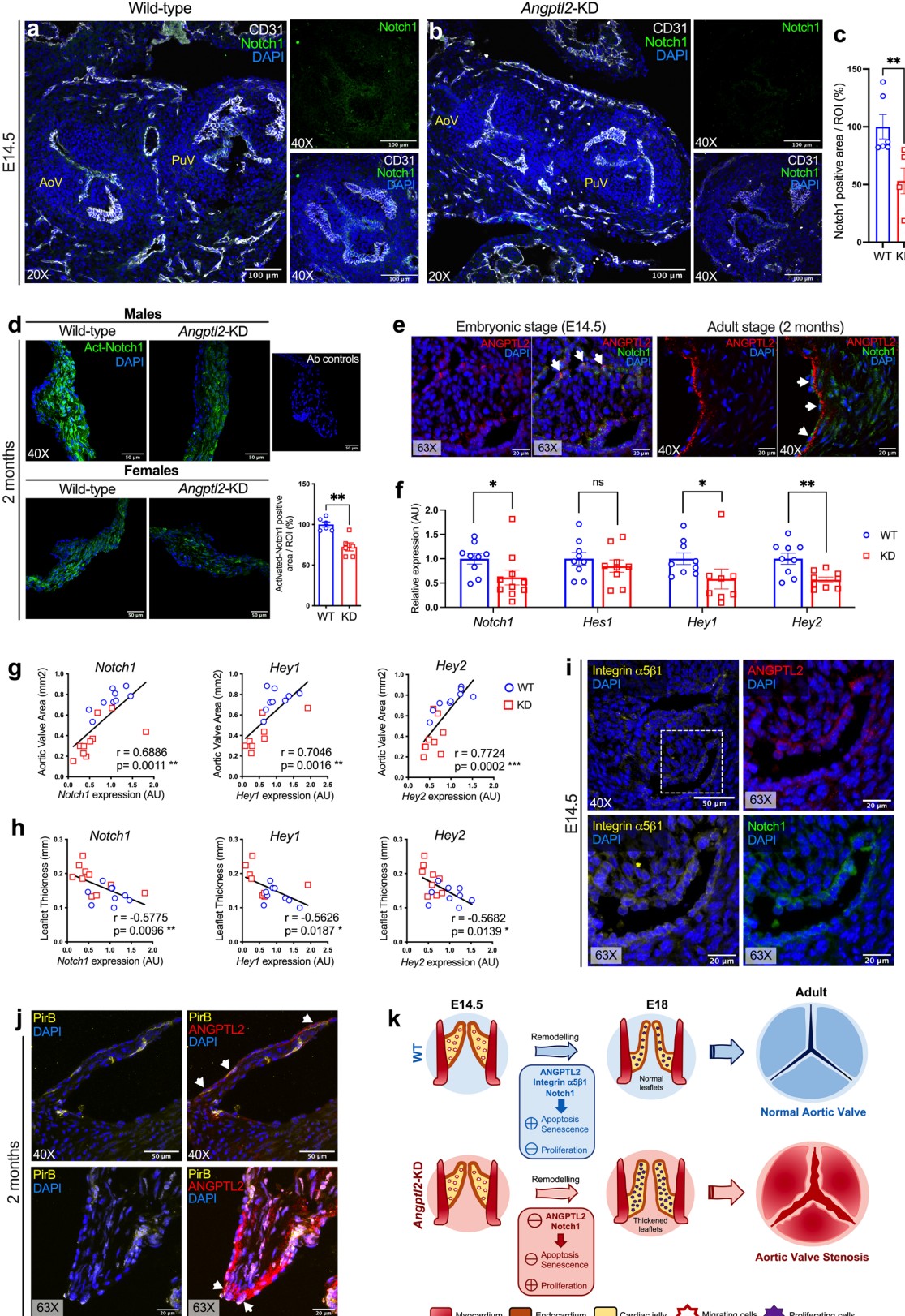

than in the regulation of EMT processes. Previous studies pointed out that *Angptl2* is expressed during vascular development in zebrafish and heart development in chicken[41,58,59], but to the best of our knowledge, this is the first study to show the specific involvement of ANGPTL2 in the development of mammalian heart valves, a complex mostly unknown process.

Embryonic programmed senescence, a crucial transient mechanism for patterning specific regions during organs and tissues development, is a relatively new concept[60], as senescence has been mostly investigated as a response to a stressor in the context of aging[61]. Embryonic senescence also occurs during heart development. For instance, Lorda-Diez *et al.* recently

**Fig. 5 AoV defects in *Angptl2*-KD mice are associated with reduced Notch1 signalling. a, b** Representative images (of $n = 3$–4 independent experiments *per* genotype) of Notch1 and CD31 protein expression by immunofluorescence in cardiac sections showing both aortic valve (AoV) and pulmonary valve (PuV); Higher magnification (×40; right panels) showing AoV from E14.5 embryos of WT and *Angptl2*-KD mice. **c** The total Notch1 positive area/ROI was quantified. Data are means ± SEM of $n = 3$ WT and $n = 4$ *Angptl2*-KD embryos (1 to 2 sections analyzed/embryo); **: $p < 0.01$ determined with Mann–Whitney U test. **d** Representative images (of $n = 6$ independent experiments) of activated Notch1 (Act-Notch1) protein expression by immunofluorescence in cardiac histological sections from adult 2-month old WT and *Angptl2*-KD male and female mice. The total Act-Notch1 positive area/ROI was quantified. Data are means ± SEM; $n = 6$ mice *per* genotype, males and females were pooled; **: $p < 0.01$ determined with Mann–Whitney U test. **e** Representative images (of $n = 3$ independent experiments) of ANGPTL2 and Notch1 protein expression by immunofluorescence in AoV from E14.5 embryos and 2-month old WT mice. Cells co-expressing ANGPTL2 and Notch1 are indicated by white arrows in AoV leaflets of WT mice. **f** Gene expression by quantitative RT-qPCR in AoV leaflets from 4-month old males and females *Angptl2*-KD mice compared to their WT littermates. Data are means ± SEM of $n = 9$ WT and $n = 8$–10 *Angptl2*-KD mice; *: $p < 0.05$ and **: $p < 0.01$ determined with t-test or Mann–Whitney U test according to the normality of distribution. **g, h** Linear correlations between *Notch1*, *Hey1* and *Hey2* mRNA levels in AoV leaflets and AoV area (**g**) and leaflet thickness (**h**) in male and female 4-month old WT and *Angptl2*-KD mice. Data are means ± SEM of $n = 17$–19 mice; *: $p < 0.05$ determined with Spearman or Pearson correlation tests according to the normality of distribution. **i** Representative images (of $n = 3$ independent experiments) of Integrin α5β1 receptor, ANGPTL2 and Notch1 protein expression by immunofluorescence in AoV from E14.5 embryo from WT mice. Higher magnification shows that Integrin α5β1, ANGPTL2 and Notch1 are expressed in the border of the leaflet. **j** Representative images (of $n = 3$ independent experiments) of PirB receptor and ANGPTL2 protein expression by immunofluorescence in AoV from 2-month old WT mice. Higher magnification shows that PirB and ANGPTL2 are co-expressed in some cells in the border of the leaflet (indicated by white arrows). DAPI staining was used to visualize cell nuclei. **k** Proposed mechanism for AVS pathogenesis in adult *Angptl2*-KD mice; At E14.5 during semilunar valve remodelling, the KD of *Angptl2* induces a decrease in Notch1 signalling via Integrin α5β1 receptors in AoV leaflets, which in turn deregulates the fine balance between cell apoptosis, senescence and proliferation. Lower apoptosis and senescence combined with higher cellular proliferation induce inadequate AoV maturation, and lead to thickened AoV leaflets at E18. These structural, cellular and molecular dysfunctions observed at embryonic stage progress to AVS in adult *Angptl2*-KD mice, in both male and female mice. **k** Was adapted from Lin et al.[87].

demonstrated the existence of intense areas of transitory senescence in the developing chicken hearts from E5 to E8, particularly in the OFT[10]. Combined with the fact that *Angptl2* is strongly expressed in the OFT of chicken hearts at E4[41], these data suggest that ANGPTL2 is involved in the regulation of a programmed senescence pathway during the development of chick AoV leaflets. Using single-cell RNA sequencing of human embryonic heart, gene expression of *ANGPTL2*, *HEY2* and *CDKN1a* (*p21*) was detected, among other genes, in the same cluster of valvar endothelial cells; gene ontology analysis suggested that these genes were associated with response to wounding and apoptosis[62]. To the best of our knowledge, our study provides new evidence for the involvement of *Angptl2* in embryonic senescence, in the development of AoV both in mice and in humans. In *Angptl2*-KD mice, reduced programmed senescence combined with lower programmed cell death and increased cellular proliferation (the higher the proliferation, the lower senescence and apoptosis), resulted in the thickening of AoV at E18, leading to AVS in adult mice.

Our data demonstrate that the expression of the Notch signalling pathway is reduced in AoV leaflets of *Angptl2*-KD mice both at embryonic and adult stages, of both sexes. The key role of Notch signalling in valve development and homeostasis has been extensively studied[63] and, accordingly, Notch deficiency is involved in congenital heart valve diseases such as BAV or acquired disorders such as calcific AoV disease[48–51]. Little is known about the potential link between ANGPTL2 and Notch signalling, but our data strongly suggest that ANGPTL2 regulates Notch-dependent remodelling processes: i) ANGPTL2 and NOTCH1 are both co-expressed in endothelial cells during AoV remodelling at E14.5 in the border of the leaflet, ii) there is a decrease of Notch signalling with a deregulation of proliferation, senescence and apoptosis in the AoV leaflets of *Angptl2*-KD mice and iii) the decrease in *Notch1*, *Hey1* and *Hey2* mRNA expressions were highly correlated to the decrease of AoV area and increase of AoV leaflet thickness, i.e., to the severity of AVS observed in *Angptl2*-KD mice. Altogether, these data strongly suggest that the tandem ANGPTL2/Notch1 is essential to AoV development and structure in mice. In agreement with our results, a direct interaction between the ANGPTL2 receptor

LILRB2 and the NOTCH1 receptor was demonstrated in HEK293T cells, and human hematopoietic stem cells stimulated with recombinant ANGPTL2 showed a rapid increase in NOTCH receptor cleavage, permitting activation of target genes[64]. This latter study established that, in accordance with our data, ANGPTL2 is not a ligand for Notch receptor, LILRB2 is not a receptor for Notch ligands, ANGPTL2 is not a target gene for Notch, but rather, ANGPTL2 acts upstream of Notch[64]. A strong genetic interaction between ANGPTL2 and NOTCH was also observed in zebrafish, as morpholino knockdown of *Angptls* 1 and 2 resulted in an absence of Notch signalling, particularly in the vasculature[65].

Post-natal interactions between NOTCH and other ANGPTLs have also been described in the literature: ANGPTL3, beyond its known effects on lipids, affects endothelial function by binding integrin α5β3 receptors in EC, stimulates the Wnt/βcatenin signalling, which in turn downregulates Notch signalling, leading to EC apoptosis and EC dysfunction[66]. A study also showed that ANGPTL4 expression was inhibited by Notch in the adult mouse heart[67]. Finally, osteogenic differentiation is accompanied by parallel up-regulation of ANGPTL-1, -2, -3 and -5 and of Notch ligands Dll1 and Dll4 expression[68].

We also show that ANGPTL2 is strongly expressed in the AoV of 2-month old mice, mostly at the border of the leaflet, where the mechanical stress is higher, suggesting that ANGPTL2 could be involved in hemodynamic resistance to high shear in physiological conditions. Indeed, it has been shown that ANGPTL2 expression is induced by mechanical stress, at least in fibroblasts[69], but further experiments are needed to determinate the specific role of ANGPTL2 in heart valve homeostasis. We believe that the strong expression of ANGPTL2 in adult AoV reflects a physiological response to mechanical forces: AoV is submitted to one of the highest shear stresses in the body. Shear stress has been estimated to range between 10 and 80 dyne/cm$^2$ on the AoV, with peak values in the range of 30–1500 dyne/cm$^2$ in the presence of stenosis[70]. These unique features may explain why AoV, but not MV or pulmonic valve, is specifically targeted by the repression of the mechanosensitive protein ANGPTL2. Interestingly, a recent study showed that in zebrafish, blood flow mechanical forces activate Notch signalling in endocardial cells

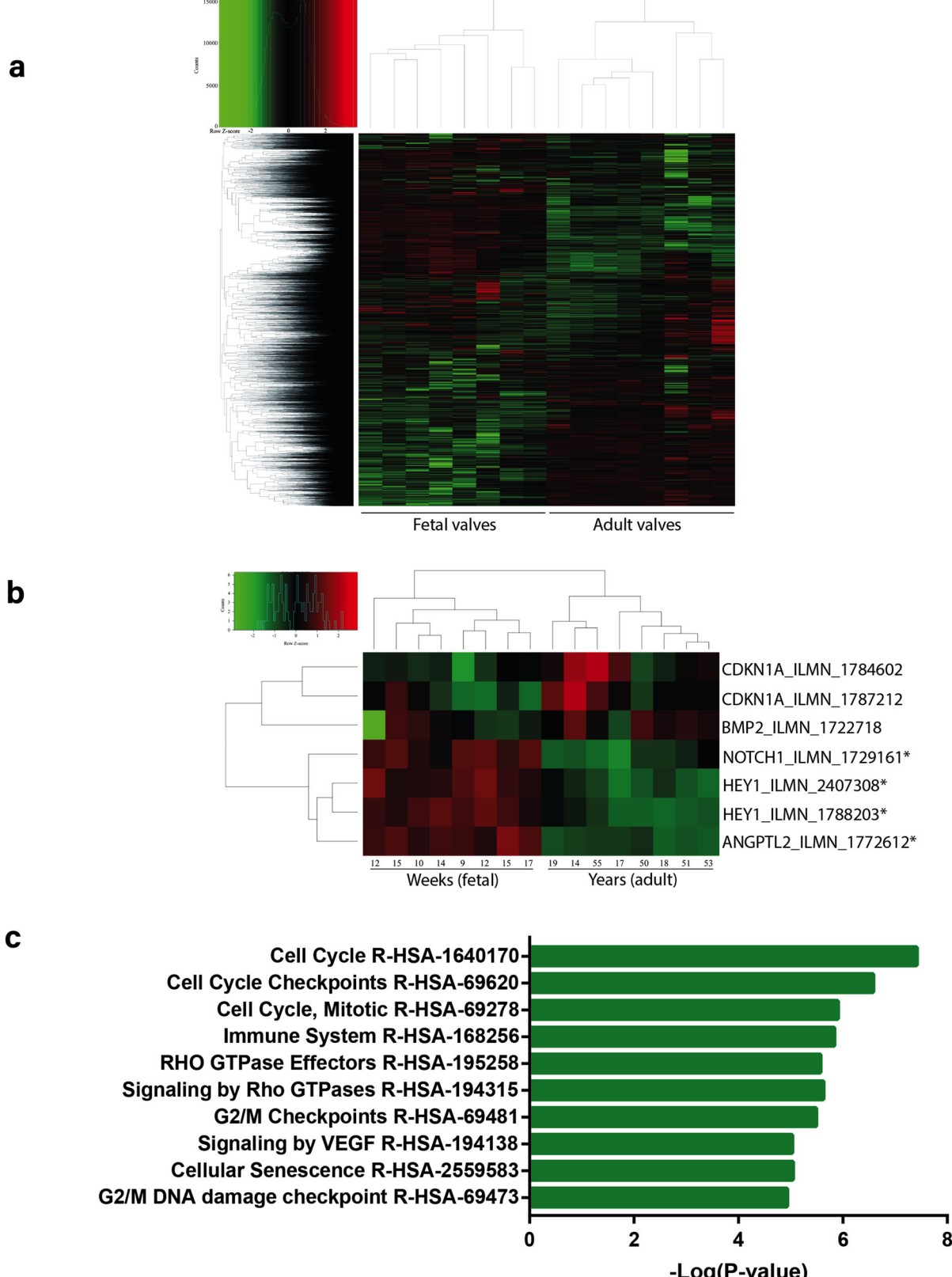

**Fig. 6 ANGPTL2 is enriched in human fetal semilunar valves and associated with cell cycle and senescence pathways. a** Heatmap of genes expressed in human fetal semilunar valves and adult AoV leaflets. **b** Close-up view of the heatmap showing significant enrichment of *ANGPTL2*, *HEY1* and *NOTCH1* in fetal semilunar valves; *significantly expressed genes (FDR < 0.05 and log2 FC > 1.5). **c** Pathway enrichment for the turquoise heart valve gene module obtained from WGCNA.

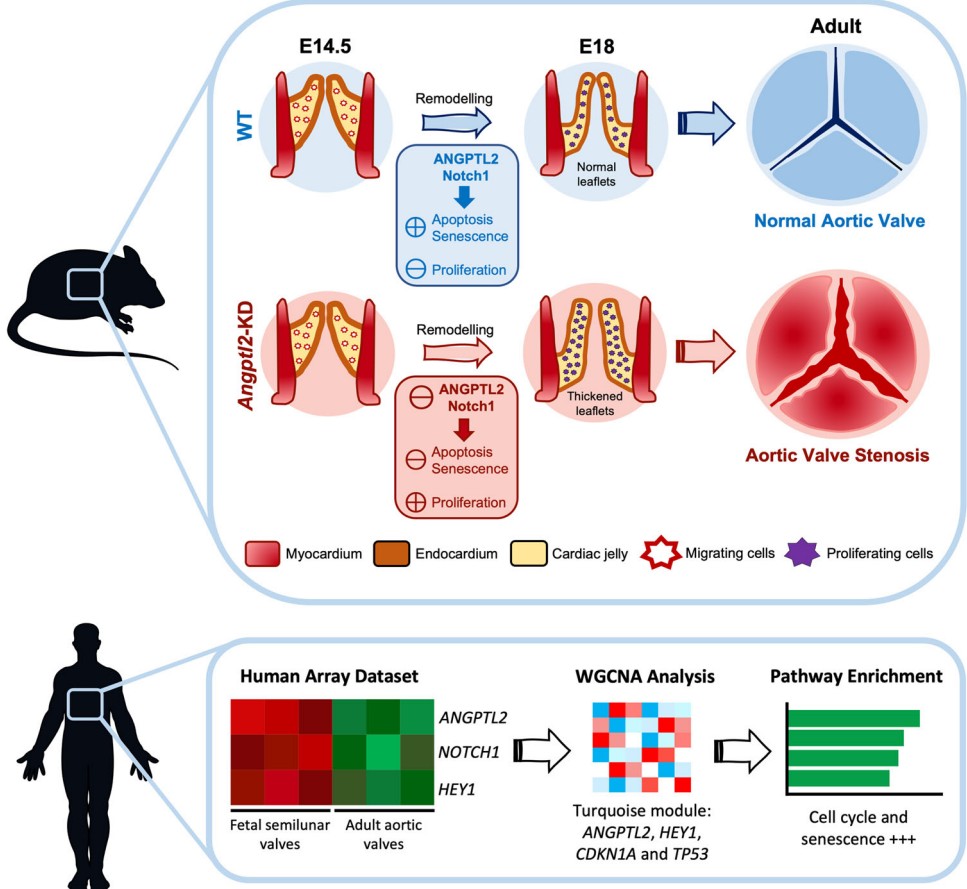

**Fig. 7 Summarized evidence for a role of ANGPTL2 as a potentially new essential player in valvulogenesis in mice and human.** At E14.5 during semilunar valve remodelling in mice, the KD of *Angptl2* induces a decrease in Notch1 signalling via Integrin α5β1 receptors in AoV leaflets, which in turn deregulates the fine balance between cell apoptosis, senescence and proliferation. Lower apoptosis and senescence combined with higher cellular proliferation induce inadequate AoV maturation, and lead to thickened AoV leaflets at E18. The thickened AoV leaflets during embryogenesis are associated with the development of AVS in adult mice. *ANGPTL2* is also enriched in human fetal semilunar valves compared to adult AoV and is associated with pathways related to cell cycle and senescence. Parts of the figure were drawn by using pictures from Servier Medical Art. Servier Medical Art by Servier is licensed under a Creative Commons Attribution 3.0 Unported License (https://creativecommons.org/licenses/by/3.0/). The upper panel was adapted from Lin et al.[87].

during valve development[71]. Blood flow forces activate Klf2/Wnt9a in Notch+ endocardial cells, leading to endocardial cells ingression into cardiac jelly, an initial essential step into valve cushions development[71]. The authors conclude that Klf2 and Notch are 2 biomechanical regulated pathways that cooperate during the sculpting of valve leaflets. Our data suggest that ANGPTL2 could be another key player in valvulogenesis in mice, adding a piece to a globally unknown process.

Spontaneous AVS observed in adult male and female *Angptl2*-KD mice was characterized by an increased leaflet thickness, collagen disorganization and remodelling. We did not observe any valve calcification, neither at 2 nor at 7 months of age under basal conditions. As mouse models do not often spontaneously calcify, dietary induction might be necessary to see more drastic changes in calcification[72]. Further experiments will be needed to investigate if there is a sexual difference in AoV leaflet fibrosis and calcification density from *Angptl2*-KD mice. Indeed, a recent study in patients with tricuspid and BAV diseases revealed that females presented a higher collagen content and less calcification density in their AoV than males[73]. At 2-month old, gene expression of *Axin2*, a downstream target and an inhibitor of Wnt/β-catenin pathway—whose loss of expression in mice delays heart valve maturation and generates immature and thickened AoV[52]—was also significantly reduced in adult *Angptl2*-KD mice.

Of note, we also observed accumulation of melanocytes in AoV leaflets from *Angptl2*-KD mice, like those reported in *Axin2*-KO mice. The origin and function of these pigmented cells are currently not known. The expression of *Bmp2*, which has been shown to promote the transition from quiescent VICs to activated VICs[74], was strongly increased, and the expression of *Bmp4*, known to be involved in OFT formation and valve remodelling[75], was decreased in *Angptl2*-KD mice. Altogether, these data suggest that ANGPTL2 contributes to valve homeostasis in mice.

We also report that adult *Angptl2*-KD mice do not develop severe LV dysfunction or heart failure. Indeed, despite a clear AVS with phenotypes ranging from aortic regurgitation, aortic valve stenosis or a combination of both, the AVS phenotype is relatively mild (grade I–II). Compared to WT, adult *Angptl2*-KD mice show the onset of a mild LV dysfunction, with reduced echocardiographic systolic and diastolic parameters, but still in the normal range, and without an increased expression of heart failure markers *Nppa* and *Nppb*. We showed that ≅70–75% of *Angptl2*-KD mice develop mild AVS and ≅ 5% exhibit a more severe AVS phenotype. The latter mice are more susceptible to the development of cardiac hypertrophy and heart failure with age. Consequently, we also report in the present study that the lifespan of *Angptl2*-KD mice is not reduced when compared to WT mice. Altogether, our data suggest that *Angptl2*-KD mice is a

model of mild-to-moderate congenital AVS that does not translate into severe cardiac dysfunction and heart failure in adults. In contrast to our data, a previous work demonstrated that *Angptl2* knockout mice show enhanced cardiac function under steady-state conditions and resistance to the development of pressure overload-induced cardiac dysfunction[76]. Discrepancies between these two mouse models may be explained by distinct genetic engineering favoring or not adaptive compensation, different substrains and age of the mice.

Analysis of a human dataset revealed that *ANGPTL2*, together with members of the Notch pathway *NOTCH1* and *HEY1*, were elevated in fetal semilunar valves compared to adult AoV. Moreover, WGCNA analysis identified *ANGPTL2*, *CDKN1A* (p21), *TP53* and *HEY1* as component of the same heart valve genetic module, and the REACTOME dataset showed their association with pathways related to cell cycle and senescence, reinforcing the hypothesis that ANGPTL2/Notch-mediated remodelling of semilunar valves can also occur in humans. Of note, it was previously reported in mice that loss of Notch signalling in VECs leads to increased proliferation of VICs accompanied by decreased expression of *Hbegf*, a negative regulator of proliferation[77]; interestingly in our study, *HBEGF* was also found with ANGPTL2 in the same module (Supplementary Data 2). WNT3A—an activator of the canonical Wnt pathway—was also part of the *ANGPTL2* gene module, which is consistent with the decrease of *Axin2* expression observed in AoV leaflets from *Angptl2*-KD mice and suggests that Wnt pathway could be also involved in ANGPTL2-mediated valve development. In the literature, *ANGPTL2* gene was detected in the single-cell transcriptome of human heart embryos in the first trimester in a cluster of fibroblast-like cells, based mainly at the base of the OFT and the valve apparatus[78]. In addition, *ANGPTL2*, together with the senescent marker *CDKN1A* and the Notch1 effector *HEY2*, were detected in the transcriptome signature in VECs and in fibroblasts-like and cardiac fibroblasts of human heart embryos[62]. Altogether, these data suggest that *ANGPTL2* contributes to the complex remodelling and signalling pathway underlying human embryonic valve development and maturation.

Our results suggest that ANGPTL2 could regulate NOTCH1 pathway *via* its receptor Integrin α5β1 in embryonic VECs, rather than *via* PirB which is not expressed in heart valves at this embryonic stage. To the best of our knowledge, little is known on the potential role of Integrin α5β1 and/or PirB/LILRB2 in heart valve development and homeostasis. Interestingly, Integrin α5β1 has been shown to be essential for the shaping of the heart valve leaflets in zebrafish[79]. By exploring the public database Gene Expression Database (GXD)[80] at the Mouse Genome Informatics website (MGI; http://www.informatics.jax.org) which provides integrated data on gene expression in mouse, we found that Itgb1 (encoding integrin subunit β1), but not Itga5 (encoding integrin subunit α5) or PirB, is expressed in AoV at E13.5 (MGI:5467139)[81] and E18.5 (MGI:6306460)[82], in the border of the leaflet, as also shown in the present study (Fig. 5i). Our data also show that in fetal human AoV, both *LILRB2* and *ITGB1*, but not *ITGA5*, were co-expressed with *ANGPTL2* and members of the NOTCH signalling in the turquoise module (Supplementary Data 2). Interestingly, RNA-sequencing previously highlighted LILRB2 as a potential gene responsible for degeneration of BAV leaflets[83], and our team previously reported that *LILRB2* expression together with *ANGPTL2* expression are upregulated in the valvular tissue of patients with calcific AVS[29]. Overall, additional experiments will be needed to decipher the potential role of Integrin α5β1 and/or LILRB2/PirB in ANGPTL2-mediated signalling during heart valve development and homeostasis, and the possible dichotomy between murine and human mechanisms.

Until now, the literature focused essentially on the deleterious pro-inflammatory, pro-oxidative and pro-senescence effects of ANGPTL2 in various age-related diseases[17–19], but the potential beneficial involvement of ANGPTL2 in development and homeostasis remains largely understudied. Here, we report an essential role for ANGPTL2 in AoV remodelling during development in mice, by regulating the crucial Notch pathway and by contributing to control the fine balance between programmed embryonic apoptosis, senescence and proliferation in valvular cells during semilunar valvulogenesis. These findings in mice are corroborated by similar results in humans, where the expression of *ANGPTL2* and *NOTCH1* is elevated in fetal semilunar valves and associated with pathways related to cell cycle and senescence. Altogether, our study identified ANGPTL2 as a potential target for the treatment of AVS, a disease with no cure.

Our study, however, has a few limitations: First, even though *Angptl2*-KD mice are an experimental model of mild-to-moderate AVS, it is difficult to make a translational comparison between our *Angptl2*-KD mouse model and human valve disease. Indeed, to the best of our knowledge, while loss of function (LoF) mutations or rare missense/nonsense variants in humans have been reported in other ANGPTL family members to be associated to hypolipidemia (ANGPTL3-4-5) and intracranial aneurysm (ANGPTL6), none has been yet identified for *ANGPTL2*. Interestingly, and in contrast with the other ANGPTLs, *ANGPTL2* shows a strong intolerance to predicted LoF variations in the human population, with a LoF observed/expected upper bound fraction score of 0.38 (gnomAD v2.1.1; https://gnomad.broadinstitute.org)[84]: this suggests that a fraction of LoF variants in ANGPTL2 might be lethal in humans at the embryonic stage, making direct comparison with our mouse model difficult.

Second, the underlying molecular mechanisms of (i) how ANGPTL2/NOTCH regulates apoptosis/proliferation/senescence processes, and (ii) how ANGPTL2 mediates activation/regulation of Notch signalling were not fully established. Here, we identified Integrin α5β1 as a potential effector for ANGPTL2-mediated signal in embryonic AoV cells, as it is co-expressed with both ANGPTL2 and NOTCH in VECs, but further experiments are needed to validate these findings. Of note, this cannot be simply validated in valve cells in culture: normal valve remodelling is a complex process, which requires spatiotemporal coordination of valve cells activity and ECM organization[7], together with the hemodynamic environment. Indeed, shear stress has been demonstrated to be involved in semilunar valves development by driving the growth of valve endothelium and the extension of the valve in the direction of blood flow[85]. Thus, additional work will be needed to address in vitro ANGPTL2-mediated signalling in AoV development in an engineered integrated environment, such as 3D hydrogels with tunable matrix stiffness to mimic the different stimuli exerted on the valve. Nonetheless, our current in vivo data are supportive of a strong regulatory role of Integrin α5β1/ANGPTL2/NOTCH1 pathway in the embryonic development of the AoV through programmed senescence, a pathway interacting with apoptotic and proliferative responses controlling cellular patterning, cell fate determination and thus proper leaflet formation.

Third, it remains unclear why the KD of *Angptl2* results in defects in AoV, but not in the pulmonic valve or MV. Differences in intrinsic factors such as distinct embryonic origin or cellular responses, and in extrinsic factors such as variable hemodynamic stress, may result in different phenotypes. The unique features of the AoV may explain why it is specifically targeted by the repression of the mechanosensitive protein ANGPTL2.

Lastly, and in accordance with the previous comment, we demonstrated that ANGPTL2 is strongly expressed in adult AoV at the edge of the leaflet, a pattern suggesting a physiological role

of ANGPTL2 in response to mechanical forces. Nevertheless, the potential role of ANGPTL2 in valve homeostasis at adult stage deserves a more thorough and detailed study.

In conclusion, we report an essential role for *Angptl2* in AoV remodelling during development in mice, by regulating the crucial Notch pathway and by contributing to control the fine tuning between programmed embryonic apoptosis, senescence and proliferation in valvular cells. *ANGPTL2*-associated senescence and *NOTCH* pathway were also detected in human normal fetal semilunar valves. At E14.5 during semilunar valve remodelling, the KD of *Angptl2* induces a decrease in Notch1 signalling in AoV leaflets, which in turn deregulates the fine balance between cell apoptosis, senescence and proliferation. Lower apoptosis and senescence combined with higher cellular proliferation induce inadequate AoV maturation, and lead to thickened AoV leaflets at E18. These structural, cellular and molecular dysfunctions observed at embryonic stage progress to mild-to-moderate AoV stenosis in adult *Angptl2*-KD mice, from 2 to 10-month old, in both male and female mice. *Angptl2*-KD mice therefore represent another mouse model of spontaneous AVS, a disease whose underlying mechanisms are ill-defined and with unmet medical need.

## Methods

**Animals**. *Male and female Angptl2*-KD mice (C57BL/6 J strain) from our colony, previously generated by an insertion of a promoter-less trapping β-geo cassette of 6500 bp into the mouse angptl2 locus, resulting in negligible levels of ANGPTL2 mRNA and protein levels in various tissues, and WT littermates were obtained and genotyped as previously described[44,45], kept under standard conditions (24 °C; 12:12 h light/dark cycle) and fed a regular diet *ad libitum*. All animal experiments were performed in accordance with the "Guide for the Care and Use of Experimental Animals of the Canadian Council on Animal Care" and were approved by the Montreal Heart Institute Animal Research Ethics Committee (ET 2017-62-03; ET 2019-62-02).

**Echocardiography**. Trans-thoracic echocardiography was performed using an i13L probe (10–14 MHz) and Vivid 7 Dimension system (GE Healthcare Ultrasound) in 2 and 7-month old mice sedated by 2% isoflurane. Two-dimensional echocardiography was used to measure LV OFT and ascending aorta dimensions. Thickness of LV anterior and posterior walls at end-diastole, LV dimensions at end-diastole and end-systole, and left atrial dimensions at end-systole were measured by M-mode echocardiography. LV mass was calculated using a previously recommended formula[86]. LV fractional shortening and EF were obtained by a formula available within the Vivid 7 system. Aortic trans-valvular peak velocity, peak and mean gradient were measured by sample volume-enlarged Doppler. Trans-mitral flow peak velocity in early filling (E), deceleration time, deceleration rate, time interval from mitral valve closure to opening were measured by pulsed-wave Doppler. LV ejection time, stroke volume and cardiac output were measured using the LV OFT flow measured by pulsed-wave Doppler. LV iso-volumetric relaxation time was measured by sample volume-enlarged pulsed-wave Doppler. The average of three consecutive cardiac cycles was used for all measurements.

**Histological studies**. Hearts (from E14.5, E18, 2- and 7-month old mice) were embedded in OCT and sectioned at 7 μm intervals. Hematoxylin and eosin, Masson's trichrome, Alcian blue, Alizarin red, Oil-Red and Picrosirius red stainings were performed and imaged with an Olympus BX45 microscope. 20× and 40× magnitude pictures were taken. All quantifications were performed using ImageJ software, version 2.0.

**Real-time qPCR**. Total RNA was extracted from dissected aortic valve leaflets (from 4- and 10-month old mice) using RNeasy mini-kit (Qiagen) and reverse transcription was performed using the Moloney murine leukemia virus reverse transcriptase (ThermoFisher) according to the manufacturer's instructions. Quantitative polymerase chain reaction (qPCR) was performed using 2X Platinum SYBR Green qPCRSuperMix-UDG (ThermoFisher) and MX3005p qPCR System cycler (Stratagene) according to the manufacturer's protocol. Primers of target genes were designed using the Genscript PCR primer design tool (Table S3). ΔΔCT were determined with MxPro v4.10 software using cyclophilin A as a housekeeping gene.

**Immunofluorescence**. Fixed tissues (transversely frozen heart sections showing aortic valve from E14.5, E18 and 2-month old mice) or cells were incubated with 1:50 diluted goat anti-mAngptl2 (AF1444, R&D Systems); 1:100 diluted rabbit anti-

CD31 (ab28364, Abcam); 1:100 diluted rat anti-CD31 (AF3628, R&D Systems); 1:100 diluted rabbit anti-p21 (ab188224, Abcam); 1:200 diluted rabbit anti-Ki67 (ab15580, Abcam); 1:200 diluted rabbit anti-Vimentin (VP-V683, Vector Laboratories); 1:100 diluted mouse anti-α-SMA (A5228, Sigma-Aldrich); 1:100 diluted rabbit anti-activated Notch1 (ab8925, Abcam); 1:100 diluted rabbit anti-Notch1 (ab52627, Abcam); 1:100 diluted rat anti-Integrin α5β1 (MAB2514, Sigma-Aldrich); 1:100 diluted rabbit anti-PirB (ab284407, Abcam); 1:500 diluted secondary antibodies Alexa fluor-647 anti-rabbit (#A31573, ThermoFisher); Alexa fluor-488 anti-rat (#A21208, ThermoFisher); Alexa fluor-555 anti-mouse (#A31570, ThermoFisher) and Alexa fluor-555 anti-goat (#A21432, ThermoFisher). TUNEL assay was performed using In Situ Cell Death Detection Kit, TMR red (Sigma-Aldrich) according to the manufacturer's protocol. DNA counterstaining was performed by incubating fixed tissues with 1:600 diluted DAPI (D1306, ThermoFisher). The catalog numbers and dilutions of all the antibodies used are listed in the Table S4. Tissue cryosections were then mounted on glass slides (Superfrost Plus, Fisher Scientific) and dried overnight. Images were acquired with a LSM 710 confocal microscope (Zeiss) using Plan APO 20X/0.8, Plan APO 40X/1.3 oil DIC and Plan APO 63X/1.4 oil DIC objectives. 40× and 63× images are maximum intensity projections created with Z-stack (0.5 μm Z-steps).

**Probe synthesis**. pBlueScript SK ( + ) vector (Addgene #212205) containing the target DNA sequence for murine *Angptl2*—subcloned using EcoRI/HindIII restriction sites between the T3 and T7 promoters—was linearized with the restriction enzyme MluI (R3198S, New England Biolabs) and purified through silica membrane spin columns (20021, Qiagen). The synthesis of a 980 bp antisense probe was then realized with 500 ng of linearized DNA incubated with DIG labelling mix (11277073910, Sigma) and T3 polymerase (M0378S, New England Biolabs). The transcription product was further treated with DNase (74204, Qiagen) to clear the probe from all residual DNA. The RNA probe was then purified through a G-50 column (CA95017-623L, Amersham) and stored at −30 °C.

**Immunoblotting**. Valve interstitial cells (VICs) were isolated from AoV leaflets explants from 2-month old WT and *Angptl2*-KD mice, grown to confluence, subcultured and lysed in Laemmli sample buffer (Bio-Rad) for 10 min at 95 °C. HEK293 cells transfected with murine ANGPTL2 were lysed and the lysate clarified by centrifugation (15,000 *rpm* for 15 min at 4 °C). Protein concentration in the supernatant was determined using the BCA Protein Assay Reagent (Bio-rad). Then, 5–10 μg of proteins were loaded and separated on 10% acrylamide SDS-PAGE gels, transferred onto nitrocellulose membranes, and probed with antibodies against murine Angptl2 (1:1000, AF1444, R&D); VE-cadherin (1:500, Santacruz, CSC-28644), Vimentin (1:500, VP-V683, Vector Laboratories), α-SMA (1:500, A5228, Sigma-Aldrich). Membranes were then probed with secondary antibodies at a dilution factor of 1:5000. Chemiluminescence was used to detect protein expression (Western Lightning Plus ECL, Perkin Elmer).

**In situ hybridization**. Mouse embryos of different stages (E9.5–E11.5), previously collected from pregnant female mice and conserved in 100% methanol, were rehydrated with decreased concentrations of cold methanol in PBST (PBS 1X, 0,1% Tween 20) solutions. Embryos were then bleached in 6% hydrogen peroxide and treated with 10 μg/mL proteinase K (PB0451, Bio Basic). The reaction was stopped with cold glycine, and the embryos were post-fixed in 4% paraformaldehyde (PFA). Mouse embryos were pre-incubated with hybridization buffers at 68 °C for 2 h, and then incubated overnight with RNA probes labelled with digoxigenin (11277073910, Sigma). Embryos were then washed several times. Non-specific sites were blocked in goat serum, BSA (BP1605-100, Fisher Scientific) (100 mg/mL) and TBST for 2 h and then anti-DIG Fab fragment (11093274910, Sigma) was added to the blockage mix overnight at 4 °C. The mouse embryos were washed several times before developing the staining with NBT (11383213001, Sigma) and BCIP (11383221001, Sigma) in the dark. When the signal was strong enough, the staining was stopped by washing the embryos in PBST and the coloration was fixed in 4% PFA.

**Gene expression in human valves**. Gene expression profiling by Illumina array was downloaded from the Gene Expression Omnibus (GSE45821). Data from 8 fetal semilunar valves (9, 10, 12, 12, 14, 15, 15 and 17 weeks) were compared to 8 adult aortic valves (14, 17, 18, 19, 50, 51, 53, 55 years old). Data from Illumina HumanRef-8 WG-DASL v3.0 BeadArray were analyzed by using mean intensity and quantile normalization. The limma workflow was used to assess differentially expressed genes (DEG). DEG were considered significant at log2 fold-change > 1.5 and FDR < 5%. NetworkAnalyst was used to asses DEG. Gene expression data was illustrated by using heatmap in R. Gene expression intensity matrix was analyzed in the R package WGCNA. The network was constructed by using consensus module detection (blockwiseConsensusModules) with default values. Gene enrichment pathway was analyzed by using the REACTOME dataset in Enrichr.

**OCT imaging**. Detailed method for OCT imaging is available in the Supplementary Methods at the end of the Supplementary Material file.

**Statistics and reproducibility**. All the data are expressed as means ± SEM. Statistical tests were performed with GraphPad Prism 9 Software for macOS according to the normality of distribution: Two-sided Student *t* tests or Mann–Whitney U tests to assess differences between two groups, and Spearman or Pearson correlation tests to measure association between two variables. A *p* value < 0.05 was considered statistically significant. Sample sizes and replicates are described in legends of figures.

**Reporting summary**. Further information on research design is available in the Nature Portfolio Reporting Summary linked to this article.

## Data availability

Gene expression profiling by Illumina array that support the findings of this study was downloaded from the Gene Expression Omnibus under accession number GSE45821[57]. In situ hybridizations on whole sagittal sections of wild-type mice embryos at E14.5 that support the findings of this study was downloaded from the public database Eurexpress under accession number euxassay_007716[46]. The pBlueScript SK ( + ) vector used in this study is available from Addgene under ID 212205. Source data are available in the Supplementary Data 1–4. Unedited blots/gels for Fig. S2 are available in the Fig. S12.

Data supporting the findings of this study are available within the paper and its supplementary information files, or are available from the corresponding author on reasonable request.

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

## Acknowledgements

This work was supported by funding from the Canadian Institute for Health Research (grants 166110 and 162446 to E.T.), the Natural Sciences and Engineering Research Council of Canada (grant RGPIN-2017-04770) and the Foundation of the Montreal Heart Institute (E.T.).

## Author contributions

P.L. designed the research, carried out experiments in mice, analyzed the data and wrote the paper; V.M.G. performed probes synthesis and in situ hybridization experiments, and wrote the paper; L.V. carried out confocal microscopy; I.D. and M.K. supervised mouse embryos dissections, probes synthesis and in situ hybridization experiments. C.D. performed embryos dissections and probes syntheses; C.M. imaged aortic valves by optical coherence tomography and carried out experiments in 7-month old mice. C.Y. supervised the analysis of the lifespan of Angptl2-KD mice. Y.F.S. performed and analyzed the trans-thoracic echocardiograms and J.C.T. validated echocardiographic data and analyses. A.A. designed the probes for in situ hybridization experiments. M.M. designed the plasmids and the primers. F.L. supervised the acquisition of aortic valves images by optical coherence tomography. P.M. and S.M. performed human valve dataset analysis and wrote the paper. N.T.-T. supervised the study design and wrote the paper; É.T. is responsible for funding the study, supervised its design and co-wrote the paper. All authors reviewed the paper.

## Competing interests
The authors declare no competing interests.
