## [Peer Review File · Communications Biology]

Reviewers' comments:

Reviewer #1 (Remarks to the Author):

Pauline LABBE et al. report that Angiotensin-like 2 affects to Aortic valve (AoV) development in mice. The authors show that the Notch1 signaling pathway is negatively regulated in Angiotensin-like2 knockdown (KD) mice, which leads to cell proliferation in the AoV and Angptl2 KD mice spontaneously develop aortic valve stenosis (AS). Overall, this study is interesting. However, additional experimental results are necessary to confirm their conclusions.

Major comments

1. In Figure 1J, the authors showed that ANGPTL2 protein is stained in the nucleus of some cells and concluded that ANGPTL2 may function as a transcription factor. However, in Figure S2F, ANGPTL2 signal was also found in nuclei of VICs from Angptl2 KD mice. We are concerned that the intranuclear ANGPTL2 signal observed in Figure 1J is non-specific signal. Authors should show any other experimental evidence. For example, authors should show the consensus sequences for binding ANGPTL2 on the transcriptional regulatory regions. Furthermore authors should show that some Angptl2 target genes are transcriptionally regulated in an ANGPTL2-dependent manner via consensus sequences..

2. There is a marked difference in the number of Ki67-positive cells between AoV of Angptl2 KD mouse shown in the Figure 3I and that shown in Figure 3J. Authors should show what makes this difference. To support the author's conclusions, it is necessary to show whether the number of Ki67-positive cells in AoV leaflets of Angptl2 KD mice correlates with the number of p21- or TUNEL-positive cells. Is the number of p21- or TUNEL-positive cells decreased in AoV of Angptl2 KD mouse shown in Figure 3I? Furthermore, are there any differences in the proliferation and/or cell death of cultured VICs from wild-type and Angptl2 KD mice?

3. In Figure 5, the authors demonstrated that Notch1 signaling is decreased in AoV of Angptl2 KD mice using immunostaining analysis of active Notch1 and gene expression analysis of Notch target genes. It is necessary to clarify the mechanism of why Angptl2 activates Notch1 signaling. The authors cited and discussed previous paper showing that ANGPTL2 activates Notch signaling in hematopoietic stem cells via LILRB2. Does the ANGPTL2 signaling through LILRB2 activate Notch1 signaling in AoV? To confirm this idea, the authors should show whether PirB, a mouse LILRB2 homolog, express in VICs.

4. In Fig. 4, H and I, the authors showed that 2-month-old Angptl2 KD mice developed AoV stenosis showing a marked pressure gradient. If the pressure gradient in the AoV is constantly present, Angptl2 KD mice would develop cardiac dysfunction and heart failure, and their lifespan will be shortened. The authors should show cardiac function evaluated by echocardiography, heart weight/body weight ratio, lung weight/body weight ratio, transcript levels of heart failure markers such as Nppa, Nppb and Myh7, and exercise tolerance by treadmill test in 1- and 1.5-year-old Angptl2 KD mice. It is essential to show data on the lifespan of Angptl2 KD mice.

5. The authors showed that Angptl2 plays an important role in the development of the AoV and that Angptl2 KD mice develop AS due to abnormal remodeling of the AoV during the embryonic period. By echocardiography, the authors showed that cardiac function of Angptl2 KD mice was slightly enhanced in 2-month-old females, but not in 2-month-old males. Moreover, the authors showed that diastolic capacity is reduced by evaluating accelerated E velocity alone (Supplemental Table S1). To support the author's conclusion, the authors should evaluate diastolic capacity by E/A, E/E', and if possible, Strain pattern.

6. The authors should show whether the other valves such as mitral valve, which has the same origin of development as the aortic valve, does not have similar proliferative changes. If the similar proliferative changes were not observed, the authors should also discuss the reason.

Reviewer #2 (Remarks to the Author):

This is a timely and elegant report by Eric Thorin's team.
The manuscript is well written and very interesting. However, the following concerns need to be addressed:

A more integrated appraisal of the relevant literature on Angiotensin-like proteins (in general) and their roles in cardiovascular disease would be appropriate to provide the context for the study.

Table S2: add info on HR, LVEDV, E/A, and cardiac output.

Figure 4I (echocardiography): please add readable scale bars for time and dimensions.

All legends should include specific "n" of biological and technical replicates for each group and a description of the statistics used for each experiment.

Primers: please add more valuable info to the supplementary table 5: for instance GenBank Accession numbers, expected size of the amplicon in bp, annealing temperatures.

Overall, the data should be discussed in greater detail in the discussion section.

The strengths and limitations of the study should be deeply addressed.

To ensure reproducibility by other laboratories, the manufacturer name and catalog number(s), concentrations used, of all the antibodies used should be listed in a supplementary table.

Minor Concern:

A proper dimensional bar should be provided for each picture.

In some IF images, DAPI is shown but not indicated.

Figure S6: please increase font size.

Point-by-point response to the reviewers

We thank the two reviewers for the opportunity to resubmit our article entitled “Angiopietin-like 2 is essential to aortic valve development in mice”. The peer-review was rigorous and constructive, acknowledging several concerns that needed to be addressed. We therefore performed additional experiments and analyses to address these concerns: the Introduction and Discussion have been enriched, Figures 1, 3, 4, 5 and 6 of the manuscript have been extensively modified with some new analyses and results, and we added 5 new Figures and 3 new Tables to the Supplementary material. We now feel that the clarity and the quality of our manuscript are much improved. However, some experiments requested by Reviewer 1 will not be performed because they would create technical and time challenges, and most importantly, we believe that they are unlikely to yield a meaningful outcome, as discussed in the rebuttal. Please find below our point-by-point responses (in black) to the reviewers’ comments (in blue). Line numbers provided refer to the revised Word/PDF version of the manuscript without tracked changes.

Reviewer #1 (Remarks to the Author):

Pauline LABBE et al. report that Angiopietin-like 2 affects to Aortic valve (AoV) development in mice. The authors show that the Notch1 signaling pathway is negatively regulated in Angiopietin-like2 knockdown (KD) mice, which leads to cell proliferation in the AoV and Angptl2 KD mice spontaneously develop aortic valve stenosis (AS). Overall, this study is interesting. However, additional experimental results are necessary to confirm their conclusions.

Major comments

1. In Figure 1J, the authors showed that ANGPTL2 protein is stained in the nucleus of some cells and concluded that ANGPTL2 may function as a transcription factor. However, in Figure S2F, ANGPTL2 signal was also found in nuclei of VICs from Angptl2 KD mice. We are concerned that the intranuclear ANGPTL2 signal observed in Figure 1J is non-specific signal.

We thank Reviewer 1 for this important comment, and we agree that the intranuclear Angptl2 signal observed in both WT and KD mice is a non-specific signal. To clarify the specificity of our antibody, we performed additional immunofluorescence experiments on E14.5 hearts sections, and repeated them on adult heart sections from *Angptl2*-KD and WT mice. The images are compiled in a new Supplemental Figure (Figure S3):

-In embryonic AoV tissue from both WT and *Angptl2*-KD mice, a systematic nuclear background signal is present (Figure S3AB-A'B'). Only in embryonic AoV tissue from WT mice, a cytoplasmic signal systematically localized on the border of the leaflets (Figure S3A; indicated by white arrowhead) is present; this cytoplasmic signal is higher than the nuclear background.

-In AoV sections from adult *Angptl2*-KD mice (Figure S3D), no signal (cytoplasmic or nuclear) was observed in the leaflet/pillar of the valve compared to WT mice (Figure S3C); only by increasing the contrast, we could see a faint non-specific background signal in some nuclei.

The difference in intensity of the non-specific background observed between E14.5 and adult valves is likely related to the difference of apparent expression levels of ANGPTL2. Indeed, confocal laser parameters for image acquisition of embryonic heart sections were 4-fold higher than those for adult heart sections, suggesting a lower ANGPTL2 expression in embryos vs. adults. High laser parameters revealed this non-specific nuclear background.

Overall, the aim of the original Figure 1 was not to compare the intensity of expression for ANGPTL2 through the different stages of development, as this evaluation is difficult considering that we assembled different experiments assessing either *Angptl2* mRNA levels or ANGPTL2 protein expression. Rather, our aim was to determine whether ANGPTL2 is expressed in mouse embryonic AoV and later in mouse adult AoV, as this had not been investigated so far. We are confident that ANGPTL2 is specifically expressed in embryonic AoV, as several evidence (ours and publicly available data) back up this statement, in mice (www.eurexpress.org; euxassay_007716), chicken (Nikki et al., 2009 PMID: 19951324) and humans (Figure 6). In fact, the low level of ANGPTL2 expression at E14.5 (when heart valve remodelling is already occurring) does not necessarily mean that ANGPTL2 is not expressed at higher levels before E14.5; we were, however, limited in the choice of experiments as i) making heart valve sections of mouse embryos before the stage of E14.5 is very challenging, as the heart becomes very small; ii) performing *in situ* hybridization on whole mouse embryo is very difficult after the stage of E12-E13, as the embryo becomes thicker and difficult to permeabilize to give access to the RNA probes, and iii) measuring *Angptl2* mRNA expression by quantitative PCR is nearly impossible due to the extremely limited amount of embryonic AoV tissue.

However, it is obvious that, according to our immunofluorescence experiments, ANGPTL2 protein is strongly expressed in 2-month old adult AoV. This is very interesting, as nothing has yet been described on the potential homeostatic role of ANGPTL2 in AoV in the literature. The AoV is submitted to one of the highest shear stresses of the body, and it has been previously described that ANGPTL2 expression is induced by mechanical stress in fibroblasts (Nakamura et al., 2014 PMID: 24465594). As mentioned in the discussion of the revised manuscript, lines 550-553 p.23: “we demonstrated that ANGPTL2 is strongly expressed in adult AoV at the edge of the leaflet, a pattern suggesting a physiological role of ANGPTL2 in response to mechanical forces. Nevertheless, the potential role of ANGPTL2 in valve homeostasis at adult stage deserves a more thorough and detailed study.”

In conclusion, and as raised by Reviewer 1, the signal observed for ANGPTL2 in some nuclei of valve cells in E14.5 heart sections of WT mice is a non-specific nuclear background observed with the antibody (AF1444; R&D). Therefore, in the revised version of the manuscript, we deleted the following sentences on lines 87-88 p.6: “At E14.5, ANGPTL2 was uniformly expressed within the leaflet of the AoV and PuV”; and on lines 90-91 p.6: “Of note, ANGPTL2 signal was also found in some cell nuclei, suggesting its potential role as a transcription factor”.

The revised version of the manuscript now states: lines 102-108 p.6, “Additionally, the spatial expression of ANGPTL2 in AoV leaflets during its development was assessed by immunofluorescence in cardiac histological sections from E14.5 and E18 WT mouse embryos. At E14.5, ANGPTL2 was expressed at the junction of the three leaflets of the PuV and the AoV (Figure 1I). Higher magnifications show that ANGPTL2 and CD31 are co-expressed in the

cytoplasm of valve endothelial cells (VECs) in both the PuV (Figure 1I; 1) and the AoV (Figure 1I; 2). At E18, ANGPTL2 expression was also restricted to the border of the leaflet in the AoV (Figure 1J)."

We also completed the paragraph on ANGPTL2 expression in VICs in the revised version of the manuscript, lines 122-126 p.7, "The cytoplasmic expression of ANGPTL2 was verified in cultured VICs from WT mice by immunofluorescence (Figure S2D). Of note, the faint nuclear staining observed in cultured VICs from adult *Angptl2*-KD mice, but also in cardiac histological sections from E14.5 WT and *Angptl2*-KD mice, is a non-specific signal (Figure S3)."

The conclusion of the paragraph has also been modified, lines 127-130 p.7, "Altogether, these results suggest that during AoV formation in WT mice, ANGPTL2 is expressed from E11 to young adult stage, with its expression restricted to the VECs at the edge of the leaflet at embryonic stage, and then extended to the VICs at the border of the leaflet at adult stage."

Authors should show any other experimental evidence. For example, authors should show the consensus sequences for binding ANGPTL2 on the transcriptional regulatory regions. Furthermore authors should show that some *Angptl2* target genes are transcriptionally regulated in an ANGPTL2-dependent manner via consensus sequences..

We totally agree with the Reviewer, even if ANGPTL2 expression was specifically detected in the nucleus – which is not the case in this study –, it does not necessarily imply that ANGPTL2 is a transcription factor. We apologize for this extrapolation and the original sentence mentioning that ANGPTL2 could be a transcription factor is now deleted.

To demonstrate that ANGPTL2 is a transcription factor is outside the scope of our present work and would require numerous studies.

2. There is a marked difference in the number of Ki67-positive cells between AoV of *Angptl2* KD mouse shown in the Figure 3I and that shown in Figure 3J. Authors should show what makes this difference. To support the author's conclusions, it is necessary to show whether the number of Ki67-positive cells in AoV leaflets of *Angptl2* KD mice correlates with the number of p21- or TUNEL-positive cells. Is the number of p21- or TUNEL-positive cells decreased in AoV of *Angptl2* KD mouse shown in Figure 3I?

We thank Reviewer 1 for this very relevant comment: it is true that senescence and apoptosis decrease in high-proliferative AoV leaflets from *Angptl2*-KD mice. We indeed observed that in AoV of *Angptl2*-KD when the proportion of Ki67-positive cells is high (original Figure 3I), the number of TUNEL+p21-positive cells is low (only representing 1% of the total nuclei count, see figure below). On the other hand, when the proportion of Ki67-positive cells is low in *Angptl2*-KD (original Figure 3J), TUNEL+p21-positive cells is higher (representing 4.7% of the total nuclei count):

At E14.5, abnormal heart valve remodelling in *Angptl2*-KD mice is therefore illustrated by an unbalance between cell proliferation, apoptosis and senescence.

To analyze potential correlations between Ki67, TUNEL and p21, the immunostaining must be performed on the same embryo and on the same (or close, within 7 μ m) heart section. For technical reasons (antibodies compatibility), immunofluorescence for p21 and TUNEL assay were performed on the same section whereas Ki67 staining was performed on a close section, from the same embryo. The thickness of a heart from a E14.5 mice embryo allows only 4-5 usable sections for each embryonic heart, limiting the number of TUNEL assay and immunofluorescence for p21 and Ki67. The number of data (n=6-8 embryos) is very low to perform correlation analyses. Nevertheless, we found:

- i) a significant negative correlation between the amount (defined as the percentage of marked cells / total nuclei count) of Ki67-positive cells and the amount of p21-positive cells ($r=-0.8032$, $p=0.0164$, $n=8$ embryos; Pearson correlation test according to the normality of the distribution evaluated by D'Agostino & Pearson test); of note, the KD embryo with high proliferation (original Figure 3I, now new Figure 3B) exhibits the lowest senescence, and on the other hand, the KD embryo with low proliferation (original Figure 3J, now new Figure 3C) exhibits high senescence (see figure below);
- ii) a tendency for a positive association between the number of TUNEL-positive cells and p21-positive cells ($r=0.7143$, $p=0.1361$, $n=6$ embryos; Spearman correlation test; n too small for normality distribution):

Correlation p21 vs. Ki67

Correlation TUNEL vs. p21

No association was found between TUNEL- and Ki67-positive cells:

Correlation TUNEL vs. Ki67

The individual data from these correlations were analyzed from the images shown in the new Figure 3 (n=1 WT and n=2 KD embryos) and from the following images (figure below, n=3 WT and n=2 KD embryos), resulting in a total of n=4 WT embryos and n=4 KD embryos:

To better illustrate heart valve remodelling and the inverse relationship between proliferation and cell death/senescence, we modified and completed Figure 3: for each AoV showed, *i.e.* WT AoV (Figure 3A), *Angptl2*-KD AoV with high proliferation (Figure 3B) and *Angptl2*-KD AoV with low proliferation (Figure 3C), we added the corresponding images of H&E staining (in Figure 3A', 3B' and 3C', respectively), immunofluorescence for Ki67 and TUNEL/p21 (in Figure 3D, 3E and 3F, respectively), and a pie chart illustrating the percentage of marked cells of the total nuclei, and the corresponding proportion (%) of TUNEL, p21 and Ki67-positive cells within the marked cells (Figure 3G, 3H and 3I, respectively).

As stated in the revised version of the manuscript, lines 153-172 p.8-9: “The analysis of TUNEL, p21 and Ki67-positive cells specifically in the AoV leaflets (Figures 3D-F) showed an imbalance between senescence/apoptosis and proliferation in *Angptl2*-KD mice compared to WT mice: pie charts illustrating the percentage of marked cells of the total nuclei, and the corresponding proportion (%) of TUNEL, p21 and Ki67-positive cells within the marked cells (Figure 3G, 3H and 3I) show i) high proliferation in *Angptl2*-KD, with heterogeneous increased number of Ki67-positive cells between AoV of *Angptl2*-KD embryos and ii) low senescence and apoptosis in AoV leaflets from *Angptl2*-KD embryos. Indeed, in AoV from *Angptl2*-KD embryo with severe valve thickening (Figure 3B'), the level of proliferation is high with 97.6% Ki67-positive cells (Figure 3E-H) and is associated with low cell death/senescence with 2.4% TUNEL+p21-positive cells of the total marked cells (Figure 3H). On the other hand, in AoV from *Angptl2*-KD embryo with thinner leaflets (Figure 3C'), the level of proliferation is moderate with 37% Ki67-positive cells (Figure 3F-I) and is associated with higher cell death/senescence with 63% TUNEL+p21-positive cells of the total marked cells (Figure 3I). In AoV from WT mice, Ki67-positive cells represent only 4% (Figure 3D-G) while TUNEL+p21-positive cells represent 96% of the total marked cells (Figure 3G). Overall, in AoV from WT embryo, the proportion of TUNEL+p21-positive cells is the highest, representing an average of 6.1% (n=4 embryos) when expressed as a % of the total nuclei number. In contrast, in AoV from *Angptl2*-KD embryo, the proportion of TUNEL+p21-positive cells is lower, representing an average of 2.8% (n=4 embryos) of the total nuclei number.” Altogether, these data suggest that abnormal heart valve remodelling in *Angptl2*-KD mice is characterized by high cellular proliferation, which is associated with low apoptosis and senescence.”

In conclusion, these new data strongly suggest that, as Reviewer 1 proposed, and as mentioned in the revised manuscript lines 182-184 p.9: “AoV leaflets from *Angptl2*-KD mice are prone to reduced apoptosis and senescence, and increased proliferation at E14.5 – the lower the senescence and apoptosis, the higher the proliferation – leading to thickened AoV leaflets at E18.”

Furthermore, are there any differences in the proliferation and/or cell death of cultured VICs from wild-type and *Angptl2* KD mice?

Unfortunately, it will not be possible to further assess the differences of proliferation/cell death in VICs cultured from WT and *Angptl2*-KD mice, for multiple reasons:

First, VICs from embryos cannot be cultured. The closest experiment would be to isolate *ex-vivo* outflow tract (OFT) from mice embryos, but it is very challenging due to its extremely small size and beyond our area of expertise.

Second, VICs cultured from adult mice would require numerous (unethical) additional 2-month old mice due to technical difficulties to explant adult valve cells: AoV leaflets from at least 4-5 mice need to be pooled to grow in one well of a 12-well plate. Moreover, and as mentioned in the discussion/limitations of the study of the revised manuscript, lines 533-540 p.23, the exploration of the underlying molecular mechanisms for ANGPTL2-mediated signalling in valve development “cannot be simply validated in valve cells in culture: normal valve remodelling is a complex process, which requires spatiotemporal coordination of valve cells activity and extracellular matrix organization (Hinton et al., 2006 PMID: 16645142), together with hemodynamic environment. Indeed, shear stress has been demonstrated to be involved in semilunar valves development, by driving the growth of valve endothelium and the extension of the valve in the direction of blood flow (Pham et al., 2022 PMID: 34535945). Thus, additional work will be needed to address *in vitro* ANGPTL2-mediated signalling in AoV development in an engineered integrated environment, such as 3D hydrogels with tunable matrix stiffness to mimic the different stimuli exerted on the valve.”

Third, and more importantly, we believe that these differences in proliferation/cell death observed at the active remodeling stage at E14.5, would not necessarily translate to the adult homeostatic stage. Indeed, when we performed TUNEL/Ki67 staining on heart sections from adult 2-month old WT and *Angptl2*-KD mice, the number of positive cells for both markers was very weak, with a lot of background and aggregates, and no significant difference was observed between WT and *Angptl2*-KD mice at that homeostatic stage (see Figure below; Apparent marked cells are indicated by yellow arrowheads; scale bar = 50 μ m).

In conclusion, cultured VICs will not be used to further characterize differences in the proliferation and/or cell death from WT and *Angptl2*-KD mice.

3. In Figure 5, the authors demonstrated that *Nothch1* signaling is decreased in AoV of *Angptl2* KD mice using immunostaining analysis of active Notch1 and gene expression analysis of Notch target genes. It is necessary to clarify the mechanism of why *Angptl2* activates Notch1 signaling. The authors cited and discussed previous paper showing that ANGPTL2 activates Notch signaling in hematopoietic stem cells via LILRB2. Does the ANGPTL2 signaling through LILRB2 activate Notch1 signaling in AoV? To confirm this idea, the authors should show whether PirB, a mouse LILRB2 homolog, express in VICs.

To the best of our knowledge, only one paper previously explored the mechanism for potential ANGPTL2-dependant Notch activation: Lin *et al.* previously showed that stimulation of human CD34⁺ progenitor cells and human endothelial cells with ANGPTL2 induces an increase in NOTCH receptor cleavage. They also further demonstrated, by performing co-IP experiments, that ANGPTL2 receptor, LILRB2, interacts with NOTCH in HEK293 cells, human CD34⁺ cells and human endothelial cells (Lin *et al.*, 2015 PMID: 25714926). Later, Horiguchi *et al.* showed that ANGPTL2 acts through PirB to activate NOTCH signaling in mouse dendritic cells (Horiguchi *et al.*, 2019 PMID: 31727773).

To take a step further in the elucidation for the potential mechanism of ANGPTL2-dependent Notch activation in valve cells, we performed additional immunofluorescence experiments, in heart valve sections from embryonic (E14.5) and adult (2-month old) WT mice, for the two recognized receptors for ANGPTL2: i) PirB, the mouse homolog for LILRB2, which was previously identified as a potential receptor for ANGPTL2 in hematopoietic stem cells (Zheng *et al.*, 2012 PMID: 22660330), platelets (Fan *et al.*, 2014 PMID: 25075127) and dendritic cells (Horiguchi *et al.*, 2019 PMID: 31727773); and ii) Integrin $\alpha 5\beta 1$, previously identified as a potential receptor for ANGPTL2 in endothelial cells (Horio *et al.*, 2014 PMID: 24526691), adipocytes (Tabata *et al.*, 2009 PMID: 19723494), cancer cells (Horiguchi *et al.*, 2014 PMID: 25287946), chondrocytes (Takano *et al.*, 2021 PMID: 31581797) and macrophages (Yugami *et al.*, 2016 PMID: 27402837). Images are compiled in the new Figure 5I-5J and in the new Figure S10.

As mentioned in the revised version of the manuscript, lines 307-318 p.14: “At E14.5, Integrin $\alpha 5\beta 1$ is expressed in the membrane of cells which are localized preferably in the border of AoV leaflet, where it is found co-expressed with ANGPTL2 and Notch1 (Figure 5I). In contrast, PirB is not expressed in embryonic AoV or in PuV (Figure S10A); PirB is expressed only in few cells with polylobed nucleus, characteristic of neutrophils, scattered across the valve leaflets (Figure S10B). In 2-month-old adult mice, however, PirB is sparsely expressed on the border of the leaflet, and is co-expressed in some cells expressing ANGPTL2 (Figure 5J). At 2 months, Integrin $\alpha 5\beta 1$ is not expressed in the leaflets (Figure S10C) but it is rather expressed in the pillar of the valve, and thus no longer co-expressed with ANGPTL2 (Figure S10D). These results suggest that embryonic ANGPTL2 signalling at E14.5 acts *via* the Integrin $\alpha 5\beta 1$ rather than *via* PirB receptors. ANGPTL2/Integrin $\alpha 5\beta 1$ are expressed together with Notch1 in VECs, at the border of the developing leaflet.”

We thus discussed these new findings in the revised manuscript, lines 485-503 p.21-22: “Our results suggest that ANGPTL2 could regulate NOTCH1 pathway *via* its receptor Integrin $\alpha 5\beta 1$ in embryonic VECs, rather than *via* PirB which is not expressed in heart valves at this embryonic stage. To the best of our knowledge, little is known on the potential role of Integrin $\alpha 5\beta 1$ and/or

PirB/LILRB2 in heart valve development and homeostasis. Interestingly, Integrin $\alpha5\beta1$ has been shown to be essential for the shaping of the heart valve leaflets in zebrafish (Gunawan et al., 2019 PMID: 30635353). By exploring the public database Mouse Genome Informatics (MGI; <http://www.informatics.jax.org>) which provides integrated data on gene expression in mouse, we found that *Itgb1* (encoding integrin subunit $\beta1$), but not *Itga5* (encoding integrin subunit $\alpha5$) or *PirB*, is expressed in AoV at E13.5 (MGI:5467139) and E18.5 (MGI:6306460; Bowen et al., 2015 PMID: 26188246), in the border of the leaflet, as also shown in the present study (Figure 5I). Our data also show that in fetal human AoV, both *LILRB2* and *ITGB1*, but not *ITGA5*, were co-expressed with *ANGPTL2* and members of the NOTCH signalling in the turquoise module (Table S3). Interestingly, RNA-sequencing previously highlighted *LILRB2* as a potential gene responsible for degeneration of bicuspid AoV (BAV) leaflets (Padang et al., 2015 PMID: 25547111), and our team previously reported that *LILRB2* expression together with *ANGPTL2* expression are up-regulated in the valvular tissue of patients with calcific AVS (Bossé et al., 2009 PMID: 20031625). Overall, additional experiments will be needed to decipher the potential role of Integrin $\alpha5\beta1$ and/or *LILRB2*/*PirB* in *ANGPTL2*-mediated signalling during heart valve development and homeostasis, and the possible dichotomy between murine and human mechanisms.”

4. In Fig. 4, H and I, the authors showed that 2-month-old *Angptl2* KD mice developed AoV stenosis showing a marked pressure gradient. If the pressure gradient in the AoV is constantly present, *Angptl2* KD mice would develop cardiac dysfunction and heart failure, and their lifespan will be shortened. The authors should show cardiac function evaluated by echocardiography, heart weight/body weight ratio, lung weight/body weight ratio, transcript levels of heart failure markers such as *Nppa*, *Nppb* and *Myh7*, and exercise tolerance by treadmill test in 1- and 1.5-year-old *Angptl2* KD mice. It is essential to show data on the lifespan of *Angptl2* KD mice.

In order to better characterize the impact of AS on cardiac dysfunction, we provide new results:

- i) Supplemental echocardiographic data in 2-month and 7-month old *Angptl2*-KD mice to further evaluate the LV systolic and diastolic (dys)function,
- ii) The transcript levels of heart failure markers *Nppa*, *Nppb* and *Myh7* in 7-month old *Angptl2*-KD mice,
- iii) The survival curve of *Angptl2*-KD mice compared to WT mice until 35 months.

We will not be able to satisfy the request concerning the need for echocardiography and exercise tolerance in 1- and 1.5-year-old *Angptl2*-KD mice. Indeed, aging is not the object of our study, which is focused on the early involvement of *ANGPTL2* in aortic valve development in mice and humans. More importantly, we believe that our *Angptl2*-KD mice is not a model of severe cardiac dysfunction and heart failure, but rather a model of mild to moderate congenital aortic valve stenosis.

Indeed, we previously assessed the lifespan of *Angptl2*-KD mice compared to WT littermates: at 1- and 1.5-year-old, *Angptl2*-KD and WT littermates had the same survival rate (these data are now included in the revised manuscript in the new Figure S7).

Several points may explain the unaffected lifespan of *Angptl2*-KD mice:

First point: Although 2-month-old *Angptl2*-KD mice clearly develop AVS with phenotypes ranging from aortic regurgitation, aortic valve stenosis or a combination of both, the phenotype is relatively mild (grade I-II) compared to previously described congenital mice models of aortic valve stenosis (grade III). Additional analyses are now included in the revised version of the manuscript to clarify and emphasize this point:

A graph showing the maximal trans-valvular aortic velocity (V_{max}) in *Angptl2*-KD mice compared to WT littermates from 2 months to 7 months (Figure 4J), together with a pie chart illustrating the distribution of phenotype severity for the AVS (Figure 4K), are shown. As stated on lines 222-227 p.11: “Moreover, the distribution of AVS phenotype severity (evaluated according to trans-valvular aortic velocity values) from 2 to 7 months of age (Figure 4J-K) shows that the stenosis observed in *Angptl2*-KD mice is mild to moderate, and did not worsen with age: before 6 months, 60% of the *Angptl2*-KD mice developed mild AVS ($V_{max} \geq 150$ cm/s) and 13% developed moderate AVS ($V_{max} \geq 300$ cm/s); after 6 months, 74% of the *Angptl2*-KD mice developed mild AVS, and 4% developed moderate AVS.”

Second point: Because *Angptl2*-KD mice develop mild to moderate AVS (Figure 4J-K), and their lifespan is not affected (Figure S7) compared to WT littermates, we did not expect severe LV dysfunction. To further evaluate if the AVS observed in 2-month-old *Angptl2*-KD mice leads to the development of LV dysfunction, we performed new echocardiographic analyses in 7-month old male *Angptl2*-KD mice and WT littermates (Table S2). In accordance with our expectations, we have stated the following in the new version of the manuscript on lines 238-242 p.11: “At 7 months, compared to male WT littermates, *Angptl2*-KD mice showed the onset of mild LV dysfunction, with the ejection fraction (EF) decreasing by 11% but remaining within the normal range (74.6 ± 1.7 vs. $66.4 \pm 3.1\%$; $p < 0.05$); there were also reductions in fractional shortening (FS; 38.2 ± 1.4 vs. $32.7 \pm 2.4\%$; $p = 0.0583$), lateral contractility (2.40 ± 0.09 vs. 1.87 ± 0.08 cm/s; $p < 0.001$) and septal contractility (2.55 ± 0.09 vs. 2.18 ± 0.08 cm/s; $p < 0.01$; Table S2).”

We then evaluated whether *Angptl2*-KD mice showed signs of cardiac hypertrophy, in a pool of 7-month old *Angptl2*-KD mice ($n=8$; 5 males and 3 females, for which we performed echocardiography and then isolated the heart for gene expression) and their WT littermates.

As mentioned in the revised version of the manuscript on lines 245-248 p.12: “We did not find significant differences in the levels of *Nppa* and *Nppb* in hearts of *Angptl2*-KD mice compared to WT littermates (data not shown), suggesting that severe LV remodelling was not occurring at 7 months, in accordance with the mild LV dysfunction observed at that age (Table S2).”

We also did not find a significant difference in the level of the hypertrophy marker *Myh7* between *Angptl2*-KD and WT mice. However, contrary to *Nppa* and *Nppb* expression, the values for cardiac *Myh7* expression in *Angptl2*-KD mice were strongly heterogeneous, with some mice expressing *Myh7* up to levels almost 20-fold higher than the others. At 7 months, we also did not find any significant differences in echocardiographic parameters related to LV hypertrophy compared to WT littermates, partly due to strong heterogeneity in the data observed for *Angptl2*-KD mice.

As stated in the revised version of the manuscript on lines 248-259 p.12: “Of note, in *Angptl2*-KD mice, the trans-valvular aortic V_{max} (Figure S6A) and mean pressure gradient (Figure S6B) were positively correlated to LV mass ($r = 0.8440$, $r = 0.8938$, respectively; $p < 0.01$), LV mass/LV

dimension at end-cardiac diastole (LVDd; $r=0.8477$, $r=0.8349$, respectively; $p<0.01$), and tended to be associated to LV mass/body weight (BW; $r=0.5984$, $r=0.6802$, respectively; $p\leq0.1171$). In other terms, in *Angptl2*-KD mice, the severity of AVS was correlated to the magnitude of LV hypertrophy. Similarly, the heart expression of the cardiac hypertrophy marker *Myh7* was negatively correlated with aortic valve area (AVA), and positively correlated with Vmax and mean pressure gradient ($r=-0.7106$, $r=0.7610$ and $r=0.8201$, respectively; $p<0.05$; Figure S6C). Moreover, *Myh7* expression tended to be positively correlated to LV mass ($r=0.7039$, $p=0.0513$) and to LV mass/LVDd ($r=0.6984$, $p=0.054$; Figure S6D), and was positively associated with LV mass/BW ($r=0.7816$; $p<0.05$; Figure S6D)."

Overall, at 7 months, the differences in LV hypertrophy and *Myh7* expression levels between *Angptl2*-KD mice can be easily explained by the distribution of AoV dysfunction (Figure 4K). As mentioned in the revised manuscript on lines 458-461 p.20: "We showed that $\cong 70-75\%$ of *Angptl2*-KD mice develop mild AVS and $\cong 5\%$ exhibit a more severe AVS phenotype. The latter mice are more susceptible to the development of cardiac hypertrophy and heart failure with age." Accordingly, the two *Angptl2*-KD mice showing the most severe AVS phenotype in Figure S6 are also those with the highest LV mass/BW and the highest cardiac *Myh7* expression.

Thus, as stated on lines 263-267 p.12 of the revised manuscript: "Taken together, these results indicate that LV function is only slightly impaired in 7-month old *Angptl2*-KD mice compared to WT littermates. Although most *Angptl2*-KD mice present mild-to-moderate AVS, approximately 5% of animals develop a more severe AoV phenotype and are more susceptible to developing cardiac hypertrophy and heart failure. Overall, the longevity of *Angptl2*-KD mice was unaltered compared to WT littermates."

Third point: We previously demonstrated that the knockdown of ANGPTL2 is beneficial by increasing resistance to stress. Compared to WT mice, adult *Angptl2*-KD mice are protected against i) the metabolic stress and vascular endothelial dysfunction induced by a high fat diet (Yu et al., 2014 PMID: 25128474); ii) cerebral endothelial dysfunction induced by angiotensin II (Yu et al., 2015 PMID: 25527773); and iii) insulin resistance and weight gain (Martel et al., 2018 PMID: 29192516). We also recently showed that the endothelial knockdown of *Angptl2* in atherosclerotic LDL^{-/-}, hApoB100^{+/+} mice decreases aortic atheroma plaque progression and promotes endothelial repair (Caland et al., 2019 PMID: 31186381)."

In conclusion, *Angptl2*-KD mice develop mild to moderate congenital aortic valve stenosis that does not translate into severe cardiac dysfunction and heart failure in adults.

5. The authors showed that *Angptl2* plays an important role in the development of the AoV and that *Angptl2* KD mice develop AS due to abnormal remodeling of the AoV during the embryonic period. By echocardiography, the authors showed that cardiac function of *Angptl2* KD mice was slightly enhanced in 2-month-old females, but not in 2-month-old males. Moreover, the authors showed that diastolic capacity is reduced by evaluating accelerated E velocity alone (Supplemental Table S1). To support the author's

conclusion, the authors should evaluate diastolic capacity by E/A, E/E', and if possible, Strain pattern.

We thank Reviewer 1 for this comment. Indeed, although the severity of AVS at 2 months was the same for males and females (Figure 4H), at the same age LV ejection fraction (66.9 ± 4.7 vs. $53.4 \pm 1.6\%$; $p < 0.05$) and fractional shortening (32.5 ± 3.4 vs. $23.4 \pm 0.9\%$; $p < 0.05$) were slightly enhanced in female *Angptl2*-KD mice compared to WT littermates, but this difference was not observed in males (Table S1). We further evaluated cardiac function in 7-month old female mice ($n=3$ for WT and *Angptl2*-KD; different mice from Table S1). LV systolic function of female *Angptl2*-KD mice was not enhanced compared to WT mice at 7 months (see below). There was even a trend ($p=0.10$) at that time point towards mild LV systolic dysfunction in female *Angptl2*-KD mice compared to WT mice, with non-significant decreases in ejection fraction (59.0 ± 0.8 vs. $69.0 \pm 6.6\%$), fractional shortening (26.0 ± 0.5 vs. $34.0 \pm 5.3\%$), and both lateral (1.8 ± 0.2 vs. 2.4 ± 0.1 cm/s) and septal (1.9 ± 0.1 vs. 2.7 ± 0.1 cm/s) contractility (Table S2):

The differences described above between female *Angptl2*-KD and WT mice were of the same magnitude as those observed in males. Furthermore, we did not observe marked differences between male and female *Angptl2*-KD mice, neither in the severity of leaflet/hinge thickening (Figure 4A-B) and AoV collagen remodelling (Figure 4E-F), nor in the severity of AVS (Figure 4H). Overall, as both male and female *Angptl2*-KD mice seem to develop mild systolic dysfunction with age, we believe that evaluating the mechanisms related to potential differences between male and female *Angptl2*-KD mice in the development of this mild cardiac dysfunction is of limited interest, and outside the scope of our present work. For more clarity, we deleted the following sentence from the revised manuscript, lines 180-182 p.10: “*In Angptl2*-KD female mice, the left ventricular ejection fraction and fractional shortening were significantly increased (Table S1).”

To properly evaluate LV diastolic function, and as requested by Reviewer 1, we added E/A and E/E' ratio, together with A velocity data, at 2 months in both males and females, and at 3.5 and 7 months in males, in the new Supplemental material (Table S1-S2) of the revised manuscript. Of note, a recurrent problem was that A velocity data were not always available, particularly in *Angptl2*-KD mice: Indeed, out of 10 mice per genotype at 2 months, the A velocity was measurable for all the WT mice, whereas it was only measurable for 5/10 *Angptl2*-KD mice.

Similarly, out of 13-14 mice per genotype at 7 months, the A velocity was measurable for 9/13 WT mice, but only for 5/14 *Angptl2*-KD mice.

In 2- and 3.5-month-old male, and in 2-month-old female *Angptl2*-KD mice, neither the E/A nor the E/E' ratios were significantly modified compared to WT littermates. Indeed, E/A ratio was decreased by 24% in 2-month-old female KD and by 13% in 3.5-month-old male *Angptl2*-KD mice compared to WT mice, but this did not reach statistical significance, partly due to the very low number of available data for A velocity as mentioned above. In conclusion, and as mentioned in the original manuscript and on lines 209-211 p.10 of the revised manuscript: "*Echocardiographic data showed no significant left ventricular systolic and diastolic dysfunction (Table S1).*"

However, at 7 months, male *Angptl2*-KD mice showed a 70% increase in E/E' ratio (43.4 ± 4.1 vs 26.2 ± 2.2 ; $p < 0.01$) compared to WT mice, indicating impaired LV diastolic function. Although the E/A ratio was not significantly altered, the global myocardial performance index (MPI), which is considered a reliable and reproducible parameter to evaluate global LV systolic and diastolic dysfunction, was also significantly increased in *Angptl2*-KD mice compared to WT littermates (54.0 ± 2.0 vs 43.2 ± 3.0 ; $p < 0.01$), which supports the E/E' ratio results.

Unfortunately, strain patterns were not available for *Angptl2*-KD mice.

Altogether, and as stated in the revised version of the manuscript on lines 242-244 p.11-12 : "Diastolic function was impaired in male *Angptl2*-KD mice compared to WT littermates, with the E/E' ratio increasing by 70% (43.4 ± 4.1 vs. 26.2 ± 2.2 ; $p < 0.01$; Table S2)."

6. The authors should show whether the other valves such as mitral valve, which has the same origin of development as the aortic valve, does not have similar proliferative changes. If the similar proliferative changes were not observed, the authors should also discuss the reason.

We thank Reviewer 1 for this comment. Although the mitral valve, which originates from the atrioventricular canal (Martin et al., 2015 PMID: 28529942), has *not* the same origin of development as that of the AoV, which arises from the cushions of the LV outflow tract, we performed additional analyses to evaluate the phenotype of the mitral valve in WT and *Angptl2*-KD mice. These results are compiled in the new Figure S4. We have stated the following on lines 214-218 p.10-11 in the revised version of the manuscript: "Of note, at the age of 2 months, although trans-mitral valve (MV) peak velocity and mean gradient were significantly higher in *Angptl2*-KD mice than in WT mice, values were all clinically normal in the 20 mice scanned (Figure S4), indicating that there was no substantial remodeling. Neither MV leaflet calcification nor MV regurgitation was detected in all mice scanned by 2D and color imaging in the apical 4-chamber view (data not shown)."

These data suggest that there are no embryonic proliferative changes in the development of the MV of *Angptl2*-KD mice.

Of note, we showed that ANGPTL2 is also expressed in the pulmonic valve (Figure 1I), which has the same origin of development as the AoV. We further focused our analyses on AoV at both the embryonic and adult stages, with confocal/histological microscope images zoomed on the AoV only. Unfortunately, echocardiographic data of the pulmonic valve were not available for the group of 20 mice scanned at 2 months (Figure 4H; Figure S4), since the project was designed for left heart study.

Considering that the lifespan of *Angptl2*-KD mice is not altered compared to WT mice, it is unlikely that MV stenosis and/or regurgitation occurred even at an advanced age, as the combination of both aortic and mitral stenosis is poorly tolerated (Unger et al., 2016 PMID: 27121305). Similarly, the combination of AVS and pulmonic valve stenosis is very uncommon (Gupta et al., 2014 PMID: 24719540).

The higher pressure and mechanical forces exerted on the AoV are likely responsible for the greater susceptibility of AoV leaflets to remodeling and degeneration following dysfunctional embryonic development. As discussed in the revised version of the manuscript, on lines 423-427 p.18-19: “AoV is submitted to one of the highest shear stresses in the body. Shear stress has been estimated to range between 10 and 80 dyne/cm² on the AoV, with peak values in the range of 30–1500 dyne/cm² in the presence of stenosis (Hasan et al., 2014 PMID: 24290137). These unique features may explain why AoV, but not MV or pulmonic valve, is specifically targeted by the repression of the mechanosensitive protein ANGPTL2.”

We have added the following sentences in the limitations of the study, on lines 545-549 p.23: “It remains unclear why the KD of *Angptl2* results in defects in AoV, but not in the pulmonic valve or MV. Differences in intrinsic factors such as distinct embryonic origin or cellular responses, and in extrinsic factors such as variable hemodynamic stress, may result in different phenotypes. The unique features of the AoV may explain why it is specifically targeted by the repression of the mechanosensitive protein ANGPTL2.”

In conclusion, the KD of *Angptl2* promotes aortic valve stenosis, with little impact on mitral valve function.

Reviewer #2 (Remarks to the Author):

This is a timely and elegant report by Eric Thorin's team.

The manuscript is well written and very interesting. However, the following concerns need to be addressed:

A more integrated appraisal of the relevant literature on Angiotensin-like proteins (in general) and their roles in cardiovascular disease would be appropriate to provide the context for the study.

We thank Reviewer 2 for this relevant comment. An overview of the present literature on the role of other angiotensin-like proteins in cardiovascular diseases is now included in the introduction of the revised manuscript, lines 35-49 p.3-4:

"Inflammation and tissue remodeling involving numerous growth factors contribute to AVS pathogenesis, and circulating growth factors have been recently proposed as potential biomarkers for AVS (Hofmanis et al., 2021 PMID: 33477548). Angiotensin like-2 (ANGPTL2) is a circulating pro-inflammatory and pro-angiogenic protein (Kim et al., 1999 PMID: 10473614), a member of the large family of angiotensin-like glycoproteins (Angptls), a family of eight (angptl1-8) members that play key roles in various biological and pathological processes. In addition to ANGPTL2, three other members of the Angptls family have been shown to be associated with cardiovascular disease (CVD): ANGPTL3, ANGPTL4 and ANGPTL8. ANGPTL3 and ANGPTL4 are main regulators of lipoprotein metabolism by inhibiting lipoprotein lipase activity, and recent reviews reported that subjects with ANGPTL3 or ANGPTL4 loss of function variants have a favorable lipid profile and a significant lower risk of CVD (Morelli et al., 2020 PMID: 31856617; Kersten 2019 30893111). ANGPTL3 and ANGPTL4 may also modulate glucose metabolism and insulin sensitivity, and recent data suggest that their loss of function variants are associated with lower odds of type 2 diabetes (Kersten 2019 PMID: 30893111). Data concerning the role of ANGPTL8 in atherosclerosis, diabetes and obesity are more conflicting, but the clinical potential of elevated plasma ANGPTL8 to predict CVD, but also metabolic syndrome and liver disease is being explored (Abu-Farha et al., 2020 PMID: 33011191)."

We also concluded, lines 71-74 p.5: "However, the role of *Angptl2* in the development of the mammalian heart and valvulogenesis is largely unknown, and to the best of our knowledge, the role of the other ANGPTLs in valve biology/disease is unidentified."

Table S2: add info on HR, LVEDV, E/A, and cardiac output.

As requested, we added data on several echocardiographic parameters at 2 months, 3.5 months and 7 months, including LVEDV, E/A and cardiac output, in the new Supplemental material (Tables S1 and S2) of the revised manuscript. These parameters were not modified in *Angptl2*-KD mice compared to WT mice, at any time point. Unfortunately, heart rate data were not available for 2-month and 3.5-month old mice; however, we previously reported that HR was similar in 4 to 5-month old *Angptl2*-KD mice and WT littermates (Yu et al., 2015 PMID 25527773). In 7-month old mice, heart rate was also not modified in *Angptl2*-KD mice compared to WT littermates (460±10 bpm vs. 455±15 bpm, n=9 per group, unpublished data).

You can also refer to our response to point 5 of Reviewer 1.

Figure 4I (echocardiography): please add readable scale bars for time and dimensions.

As requested, we have added readable scale bars.

All legends should include specific "n" of biological and technical replicates for each group and a description of the statistics used for each experiment.

We apologize for the missing information. All legends of the revised manuscript now include specific *n* of embryos/mice/cardiac sections, the number of replicates for each experiment performed and a short description of the statistic used.

Primers: please add more valuable info to the supplementary table 5: for instance GenBank Accession numbers, expected size of the amplicon in bp, annealing temperatures.

We thank Reviewer 2 for this comment. For more clarity, the information on the GenBank Accession numbers for the appropriate target mRNA, the expected size of the amplicon and the annealing temperatures for all the couples of primers have been included in the new Table S6 in the revised manuscript (previous Table S5 in the original manuscript).

Overall, the data should be discussed in greater detail in the discussion section.

We thank Reviewer 2 for this comment. The discussion is now enriched with the following points integrated in the revised version of the manuscript:

- 1) Lines 485-503 p.21: the finding that Integrin $\alpha 5\beta 1$ is the potential receptor for ANGPTL2-mediated signalling during heart valve development;
- 2) Lines 388-416 p.17-18: the link between ANGPTL2 and NOTCH1 and a short overview of the present literature on the interaction between ANGPTL2 (and other Angptls), and NOTCH;
- 3) Lines 417-434 p.18-19: the fact that both ANGPTL2 and NOTCH are mechanosensitive;
- 4) Lines 453-464 p.20: the mild severity of AVS in adult *Angptl2*-KD mice, its relative low impact on LV dysfunction and the unaffected longevity of *Angptl2*-KD mice;
- 5) Lines 422-427 p.18-19; 545-549 p.23: the possible explanations for the absence of effects of ANGPTL2 knockdown in the other heart valves.

The strengths and limitations of the study should be deeply addressed.

We thank Reviewer 2 for this relevant comment. We now clearly and deeply address both the strengths and the limitations of our study in the revised version of the manuscript, at the end of the discussion lines 504-553 p.22-23.

To ensure reproducibility by other laboratories, the manufacturer name and catalog number(s), concentrations used, of all the antibodies used should be listed in a supplementary table.

As requested, the information for all the primary and secondary antibodies used are now indicated in the new Table S7.

Minor Concern:

A proper dimensional bar should be provided for each picture.

As requested, a dimensional bar is now provided for each picture.

In some IF images, DAPI is shown but not indicated.

Thank you for this comment. DAPI is now indicated in all the main IF images. In Figure 1 and Figure 2, we did not indicate DAPI in some smaller images, which are zooms of the main images, for readability. However, in all the legends of the Figures showing IF experiments, we indicated the following sentence, for clarity: “DAPI staining was used to visualize cell nuclei.”

Figure S6: please increase font size.

This has been done, as requested.

REVIEWERS' COMMENTS:

Reviewer #1 (Remarks to the Author):

The manuscript is now significantly improved and the authors answered correctly most of the comments.

However, I still have a minor concern about the author's conclusions.

In the revision, the authors concluded that 7-month-old Angptl2-KD mice show slightly impaired LV function, and that most Angptl2-KD mice develop mild to moderate congenital aortic valve steatosis, while ~5% of animals show more severe AoV phenotype and are more susceptible to developing cardiac hypertrophy and heart failure. On the other hand, a previous study has reported that Angptl2 KO mice show enhanced cardiac function under steady-state conditions and are resistant to the development of pressure overload-induced cardiac dysfunction (Tian et al., Nat Commun 7: 13016, 2016). How do the authors explain the discrepancy in phenotypes between Angptl2-KD and Angptl2 KO mice? The authors at least should mention and discuss about this issue.

Reviewer #2 (Remarks to the Author):

The Authors did a great job in revising their manuscript.

Response to the reviewers

Reviewer #1 (Remarks to the Author):

The manuscript is now significantly improved and the authors answered correctly most of the comments.

We thank Reviewer 1 for his very constructive revision of our manuscript.

However, I still have a minor concern about the author's conclusions.

In the revision, the authors concluded that 7-month-old *Angptl2*-KD mice show slightly impaired LV function, and that most *Angptl2*-KD mice develop mild to moderate congenital aortic valve steatosis, while ~5% of animals show more severe AoV phenotype and are more susceptible to developing cardiac hypertrophy and heart failure. On the other hand, a previous study has reported that *Angptl2* KO mice show enhanced cardiac function under steady-state conditions and are resistant to the development of pressure overload-induced cardiac dysfunction (Tian et al., Nat Commun 7: 13016, 2016). How do the authors explain the discrepancy in phenotypes between *Angptl2*-KD and *Angptl2* KO mice? The authors at least should mention and discuss about this issue.

We thank Reviewer 1 for this important comment. Indeed, the discrepancy in cardiac phenotypes between our *Angptl2*-KD mouse model and the *Angptl2* KO mouse model from Oike's group (Tian et al., Nat Commun. 2016;7:13016) is puzzling. We only can speculate on the origin of these differences:

1. As described in the Methods section, lines 571-573 p.25, our *Angptl2*-KD mouse model has been generated "by an insertion of a promoter-less trapping β -geo cassette of 6500 bp into the mouse *angptl2* locus, resulting in negligible [detected] levels of *ANGPTL2* mRNA and protein levels in various tissues". Therefore, background expression of *ANGPTL2* in our *Angptl2*-KD mice could be sufficient to induce activation of molecular/cellular physiological responses and induction of compensatory mechanisms, leading to more variable phenotypes (large spectrum of AVS severity) than in knockout mice where the expression of *ANGPTL2* is totally abrogated.

2. To the best of our knowledge, there are no available information on the embryonic phenotype of *Angptl2* KO mice compared to WT littermates – only a mention that they are "grossly normal" at birth (Tabata et al., Cell Metab. 2009;10(3):178-188; Tian et al., Nat Commun. 2016;7:13016). In the paper of Tian et al., cardiac function of *Angptl2* KO mice was explored at a relatively young age, 6-, 12- and 20-week-old. To the best of our knowledge, data on both cardiac function and longevity in older *Angptl2* KO mice are not available. These differences make the comparison between the two studies difficult to interpret.

3. Finally, *Angptl2*-KD and *Angptl2* KO mouse models appears to have been generated on different substrains (C57BL/6NJcl vs. C57BL/6J); genetic differences have been reported among B6 mouse strains as a result of naturally occurring genetic drift (Mekada et al., Exp Anim.

2009;58(2):141-149), resulting in significant phenotypic differences in a number of physiological, biochemical and behavioral systems (Simon et al., Genome Biol. 2013;14(7):R82).

We stated in the revised Discussion of the manuscript, lines 462-467 p.20: “In contrast to our data, a previous work demonstrated that Angptl2 knockout mice show enhanced cardiac function under steady-state conditions and resistance to the development of pressure overload-induced cardiac dysfunction (Tian et al., Nat Commun. 2016;7:13016). Discrepancies between these two mouse models may be explained by distinct genetic engineering favoring or not adaptive compensation, different substrains and age of the mice.”

Reviewer #2 (Remarks to the Author):

The Authors did a great job in revising their manuscript.

We thank Reviewer 2 for his very constructive revision of our manuscript.